# Universal in situ oxide-based ABX$_3$-structured seeds for templating halide perovskite growth in All-perovskite tandems

Weiqing Chen[1,4], Shun Zhou[1,4], Hongsen Cui[1,4], Weiwei Meng [2,4], Hongling Guan[1], Guojun Zeng[1], Yansong Ge[1], Sengke Cheng[1], Zixi Yu[1], Dexin Pu[1], Lishuai Huang[1], Jin Zhou[1], Guoyi Chen [1], Guang Li[1], Hongyi Fang[1], Zhiqiu Yu[1], Hai Zhou [3], Guojia Fang [1] & Weijun Ke [1] ✉

Precise control over halide perovskite crystallization is pivotal for realizing efficient solar cells. Here, we introduce a strategy utilizing in-situ-formed oxide-based ABX$_3$-structured seeds to regulate perovskite crystallization and growth. Introducing potassium stannate into perovskite precursors triggers a spontaneous reaction with lead iodide, producing potassium iodide and lead stannate. Potassium iodide effectively passivates defects, while PbSnO$_3$ (ABX$_3$-structured), exhibiting a 98% lattice match, acts as a template and seed. This approach facilitates pre-nucleation cluster formation, preferential grain orientation, and the elimination of intermediate-phase processes in perovskite films. Incorporating potassium stannate into both the perovskite precursors and the buried hole transport layers enables single-junction 1.25 eV-bandgap Sn-Pb perovskite solar cells to achieve a steady-state efficiency of 23.12% and enhanced stability. Furthermore, all-perovskite tandem devices yield efficiencies of 28.12% (two-terminal) and 28.81% (four-terminal). This versatile templating method also boosts the performance of 1.77 eV and 1.54 eV-bandgap cells, underscoring its broad applicability.

Metal halide perovskites with an ABX$_3$ structure have emerged as highly promising materials for thin-film solar cells, offering high power conversion efficiencies (PCEs) and low production costs[1–6]. Over the past decades, the efficiency of lead-based perovskite solar cells (PSCs) has surged from 3.8% to 26.7%, nearing that of conventional silicon solar cells[7–10]. Achieving high PCEs in PSCs relies on the development of well-crystallized perovskite absorption layers capable of efficiently capturing light while minimizing carrier recombination[11–13]. However, uncontrollable nucleation, random grain orientation, abundant surface trap states, and significant interfacial stress in polycrystalline perovskite films can degrade photovoltaic performance, underscoring the importance of controlling

perovskite crystal growth and film quality[14–16]. These common challenges become even more pronounced in all-perovskite tandem solar cells, which use mixed tin-lead (Sn-Pb) perovskites as the bottom subcells[13,17–23]. Traditional Sn-Pb perovskite films are often prepared using a one-step anti-solvent solution method[24,25], which can lead to randomly oriented polycrystalline films due to the rapid accumulation of perovskite building blocks during solvent evaporation[26]. In addition, mixed Sn-Pb perovskites often suffer from differing crystallization rates between Sn and Pb, leading to an excessively rapid and highly uncontrolled crystallization process, which complicates achieving high-quality films[24,27]. To address these challenges, regulating the nucleation and crystallization processes in Sn-Pb PSCs is

[1]Key Laboratory of Artificial Micro- and Nano-structures of Ministry of Education of China, School of Physics and Technology, Wuhan University, Wuhan, China. [2]South China Academy of Advanced Optoelectronics, South China Normal University, Guangzhou, China. [3]International School of Microelectronics, Dongguan University of Technology, Dongguan, Guangdong, China. [4]These authors contributed equally: Weiqing Chen, Shun Zhou, Hongsen Cui, Weiwei Meng. ✉e-mail: weijun.ke@whu.edu.cn

essential for balancing the crystallization rates, achieving high-orientation film growth, and minimizing surface defects and stresses[28]. The complex crystallization process of Sn-Pb perovskites differs from that of single-phase perovskites due to the coexistence of both Sn-based and Pb-based phases in mixed precursor solutions[29–31]. This adds further complexity to controlling crystal formation.

Seed-induced crystallization has proven effective in achieving uniform seeding and orientation-induced growth by reducing the critical Gibbs free energy of nucleation and increasing the nucleation rate[32–35]. Recently, this strategy has been applied to pure Pb-based PSCs with promising results. For instance, Luo et al.[36] introduced a two-dimensional (2D) perovskite seed layer to induce the epitaxial orientation in three-dimensional (3D) perovskites, achieving high-quality mixed-dimensional Pb-based perovskite films with PCEs up to 23.95%. Similarly, Zhang et al.[37] employed a heterogeneous seed-induced crystallization strategy that lowered the nucleation barrier by forming a low-solubility complex with lead iodide ($PbI_2$), resulting in a PCE of 24%. In addition, Zhao et al.[38] used an in-situ grown 2D perovskite seed layer to create a 2D/3D heterojunction with a preferred orientation, yielding PSCs with an efficiency of 24.83%. These studies demonstrated that seed crystal-induced crystallization can promote the hetero-epitaxial growth of perovskites with preferred crystal plane orientation, reduce intrinsic trap states, and enhance charge transport[39,40]. Despite these advancements, there are limited reports on seed-induced oriented crystallization in Sn-Pb perovskite films. Evidence suggests a strong correlation between defect states and crystal orientation in polycrystalline perovskite films, underscoring the need for further research in this area[41,42]. Moreover, to induce preferred orientation crystallization, seed materials must have a complete crystal structure, high phase purity, a chemical composition similar to the desired perovskite phases, and a high lattice match with perovskite crystals[37,43–46]. While current seed methods typically use halide perovskites as template seeds, which can be destabilized by solvents during crystal growth, there is a lack of focus on using stable oxide-based $ABX_3$ materials. Therefore, developing a stable seed material with high phase purity, crystallinity, and precise lattice matching is crucial for inducing oriented crystallization, minimizing defects in Sn-Pb perovskite films, and achieving high PCEs with enhanced stability in all-perovskite tandem solar cells.

Here, we developed an in-situ method for constructing oxide-based $ABX_3$ structure template seeds to induce preferred orientation crystallization in Sn-Pb perovskite films. This approach effectively addresses the challenges of random grain orientation and buried interfacial defects and stresses caused by uncontrolled crystallization rates. By introducing a small amount of potassium stannate ($K_2SnO_3$) into the perovskite precursor solutions, a spontaneous reaction with $PbI_2$ resulted in the formation of potassium iodide (KI) and seed lead stannate ($PbSnO_3$) crystals with an $ABX_3$ structure. The high adsorption priority and lattice matching between $PbSnO_3$ seeds and Sn-Pb perovskites facilitated the elimination of intermediate-phase formation and promoted preferred nucleation and rapid crystallization of Sn-Pb perovskites, enabling the growth of high-quality, uniformly crystalline Sn-Pb perovskite films. In addition, the multi-electron donor stannate and KI byproducts effectively passivated defects and inhibited ion migration. By incorporating $K_2SnO_3$ into both the perovskite precursors and the buried hole transport layers, single-junction Sn-Pb PSCs achieved an optimal steady-state efficiency of 23.12% with improved stability. Furthermore, when applied to two-terminal (2 T) and four-terminal (4 T) all-perovskite tandem solar cells, the devices achieved steady-state efficiencies of 28.12% and 28.81%, respectively. This strategy was also highly effective for 1.77 eV wide-bandgap (WBG) and 1.54 eV normal-bandgap systems, significantly improving the performance of their solar cells and demonstrating the broad applicability of this approach for perovskite materials.

## Results

### In situ construction of $ABX_3$ seed framework

In this study, $K_2SnO_3$ was introduced into the perovskite precursor solutions to facilitate the formation of inorganic oxide-based $ABX_3$ seeds. Upon reacting with $PbI_2$, $K_2SnO_3$ generated KI and $PbSnO_3$ crystals possessing an $ABX_3$ structure, as illustrated in Eq. 1.

$$K_2SnO_3 + PbI_2 \rightarrow 2KI + PbSnO_3 \qquad (1)$$

The $PbSnO_3$ crystals served as the nucleating seeds that induced the crystallization of Sn-Pb perovskites. Precursor ions then gathered around the $PbSnO_3$ seeds to form tiny crystalline nuclei. The formation of these nuclei strongly depends on the critical nucleus radius ($r^*$): if the nucleus radius is smaller than $r^*$, it will dissolve back into the solution; If the radius is larger than $r^*$, the nucleus becomes thermodynamically stable and can grow further[47]. Under the influence of $PbSnO_3$ seeds, perovskites entered the pre-nucleation stage (Fig. 1a). The seeds significantly reduced the interfacial energy of perovskites, thus lowering the nucleation barrier (Supplementary Fig. 1)[36]. As the solvent gradually evaporated, Sn-Pb perovskites nucleated uniformly and crystallized preferentially under the guidance of the $ABX_3$-structured seeds.

To verify the chemical reaction in the precursor solutions, we characterized X-ray diffraction (XRD) patterns of the resulting reaction products (Fig. 1b). After mixing $K_2SnO_3$ with $PbI_2$, XRD patterns corresponding to KI and $PbSnO_3$ were observed. In addition, diffraction peaks of $PbSnO_3$ were present when $K_2SnO_3$ was added into perovskite precursor solutions. To demonstrate the necessary conditions for the formation of $PbSnO_3$ from $PbI_2$ and $K_2SnO_3$, we also reacted $K_2SnO_3$ with other precursor materials (formamidinium iodide (FAI), methylammonium iodide (MAI), and tin (II) iodide ($SnI_2$)), as shown in Supplementary Fig. 2. However, we only detected the diffraction peaks of KI crystals in these mixtures. Furthermore, to investigate the regulatory effects of $K_2SnO_3$ on the interactions among perovskite precursors, we theoretically evaluated the reaction priorities between the corresponding organic and inorganic components, as shown in Fig. 1c. The interaction between $PbI_2$ and $K_2SnO_3$ results in the formation of $PbSnO_3$ and KI with a reaction energy of $-0.96$ eV. For comparison, the interactions between $PbI_2$ interacts with dimethyl sulfoxide (DMSO) and N, N-dimethylformamide (DMF) yield reaction energies of $-0.87$ eV ($PbI_2$•DMSO) and $-0.77$ eV ($PbI_2$•DMF), respectively. This indicates a preference for the crystallization of $PbSnO_3$ over solvent-related intermediate phases, consistent with the experimental observation of $PbSnO_3$ XRD patterns in Fig. 1b.

To further demonstrate the feasibility of this chemical reaction, we characterized the interactions between the precursor feedstock and $K_2SnO_3$ using a Fourier Transform Infrared (FTIR) Spectrometer. With the introduction of $SnI_2$ and $PbI_2$ into $K_2SnO_3$, the vibration peaks of $SnO_3^{2-}$ shifted from 693 cm$^{-1}$ to 690 cm$^{-1}$ and 687 cm$^{-1}$ (Supplementary Fig. 3), respectively. This shift is likely related to the coordination between $SnO_3^{2-}$ and $Pb^{2+}$ or $Sn^{2+}$. In addition, we characterized the colloidal size in the precursor solutions using dynamic light scattering (DLS) measurements. As shown in Fig. 1d, the introduction of $K_2SnO_3$ resulted in a significant increase in colloidal size, indicating the formation of larger atomic clusters in the precursor solutions, which is conducive to the formation of critical nuclei[37].

### Characterization of mixed Sn-Pb perovskite films

To investigate whether the in situ formation of $ABX_3$-structured seeds facilitates the preferred orientation crystallization of perovskite films, we analyzed the crystal structure and morphology of annealed films. As shown in Fig. 2a, b, films developed via seed-induced nucleation exhibited superior crystallinity and orientation compared to control films. The diffraction peak intensities for the (100) and (200) crystal planes were notably enhanced, while the intensity for the (111) crystal

 

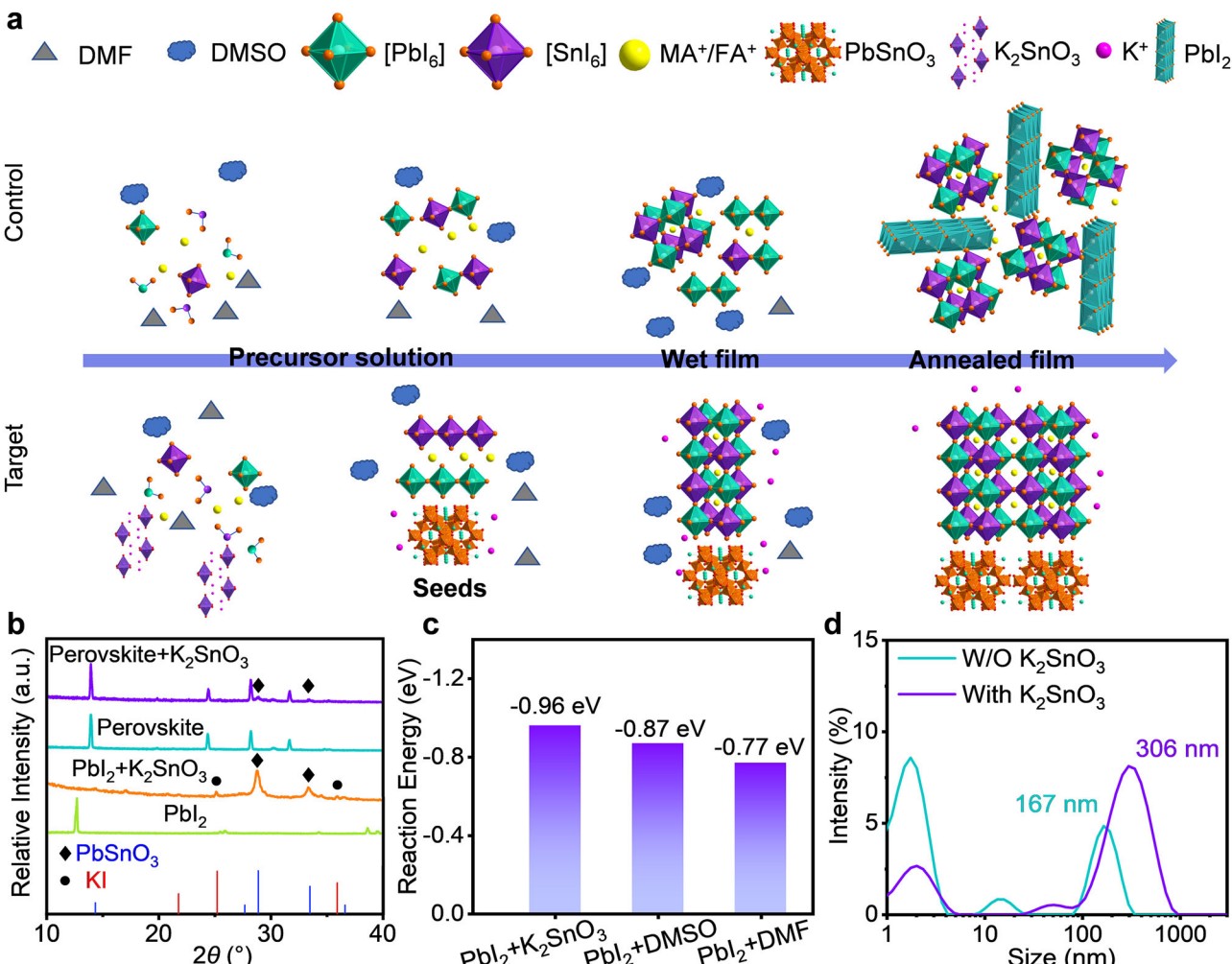

**Fig. 1 | In situ construction of ABX₃ seed framework and regulate nucleation and crystallization. a** Schematic diagram illustrating the seed-induced oriented crystallization of K₂SnO₃-treated FA₀.₇MA₀.₃Pb₀.₅Sn₀.₅I₃ perovskite films. **b** XRD patterns of PbI₂, the products following the reaction of K₂SnO₃ with PbI₂, and perovskites treated without and with K₂SnO₃ (The square symbol represents PbSnO₃:

PDF 17-0607, the circular symbol represents KI: PDF 04-0471). **c** DFT calculated reaction energies for the formation of PbSnO₃ (K₂SnO₃ + PbI₂ → 2KI + PbSnO₃), PbI₂·DMSO (DMSO + PbI₂ → PbI₂·DMSO), and PbI₂·DMF (DMF + PbI₂ → PbI₂·DMF) respectively. **d** Dynamic light scattering spectra of perovskite precursors with and without K₂SnO₃.

plane was diminished. Figure 2a also shows that no shifts were observed after adding K₂SnO₃, confirming that K₂SnO₃ does not participate in the lattice composition of the perovskites. Furthermore, comparing the full width at half maximum (FWHM) of the diffraction peaks (Supplementary Fig. 4) revealed that the target group exhibited smaller FWHM compared to the control group, indicative of improved crystallinity. Notably, the (100) and (200) crystal planes displayed enhanced orientation, which can be attributed to the role of PbSnO₃ as a seed material promoting preferred orientation crystallization of the perovskite and improving the crystal quality of the films. In addition, the pole figure along the (100) plane (Fig. 2c) revealed a marked improvement in crystal surface orientation, favoring vertical film growth relative to the substrates. This orientation is attributed to the induced crystallization by ABX₃-structured PbSnO₃ seeds. Furthermore, XRD analysis of the wet films after spin coating revealed XRD patterns corresponding to both intermediate and perovskite phases in the control group, which can lead to the formation of heterogeneous nucleation sites and cause component aggregation and crystallization (Supplementary Fig. 5)[48]. In contrast, the target films displayed only perovskite, KI and PbSnO₃ diffraction peaks, thereby eliminating the need for intermediate-phase processes. The angular diffraction patterns (Supplementary Fig. 6) of perovskite films treated with K₂SnO₃ were also measured by grazing incident X-ray diffraction (GIXRD). As

the incidence angle (ω) gradually increased, the detection depth deepened. When ω = 1.5°, the diffraction peak of PbSnO₃ was observed. As ω further increased to 2°, the diffraction peak of the indium tin oxide (ITO) substrate became evident, indicating that the ITO glass substrate was being detected. At this point, the diffraction signal of PbSnO₃ was further enhanced, and the ratio of the integrated peak areas of the PbSnO₃ to the (100) perovskite increased. These results confirm that PbSnO₃ was more concentrated near the bottom interface of the final film. Moreover, using the Bragg equation, we calculated the lattice matching rates based on the crystal plane parameters. As shown in Supplementary Table 1, the lattice matching rates between the perovskite (100) and (200) planes and PbSnO₃ are 97.80% and 98.16%, respectively. This high degree of lattice matching promoted more orderly periodic growth of halide perovskites, which was conducive to its preferred orientation growth[49].

High lattice matching contributed to the periodic orientation growth of Sn-Pb perovskite crystals and effectively relieved stress. GIXRD patterns (Supplementary Figs. 7, 8) demonstrated that the residual stresses in Sn-Pb the perovskite films were significantly reduced with the introduction of PbSnO₃ seeds. This reduction was attributed to the in situ generation of PbSnO₃ seeds within the perovskite precursors, which induced the preferred orientation crystallization of the films, preventing stress accumulation associated with

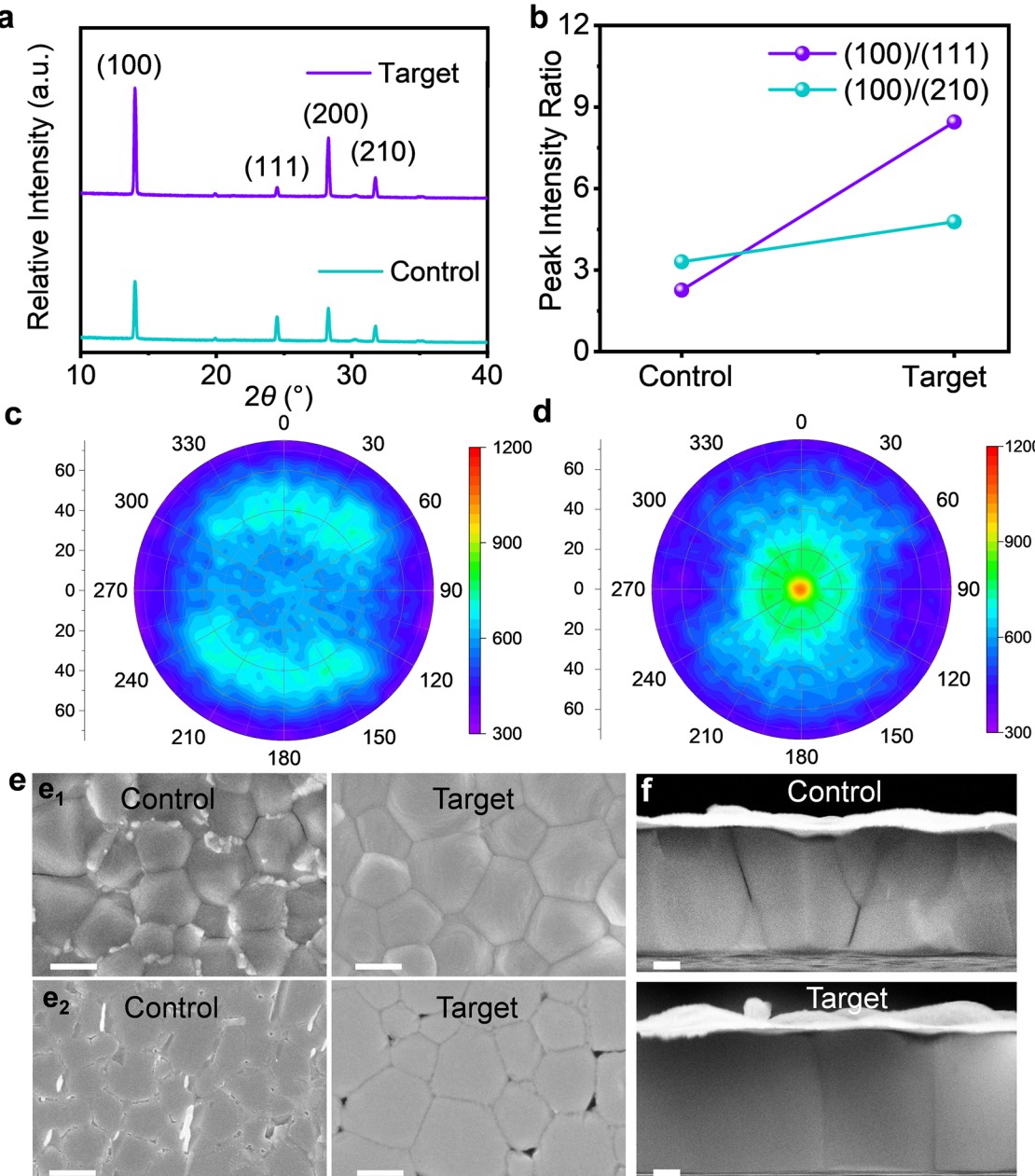

**Fig. 2 | Characterizations of mixed Sn-Pb perovskite films. a, b** XRD patterns of $K_2SnO_3$-treated $FA_{0.7}MA_{0.3}Pb_{0.5}Sn_{0.5}I_3$ perovskite films, and (**b**) the ratio statistical spectra of diffraction peak intensity to the relative crystal plane. **c, d** Pole figures of (**c**) control and (**d**) target Sn-Pb perovskite films at $2\theta = 14°$. **e** SEM images of (**e₁**) the top and (**e₂**) bottom surfaces of the control and target Sn-Pb perovskite films. The scale bar is 600 nm. **f** Cross-sectional SEM images of Sn-Pb perovskite films without and with $K_2SnO_3$ treatment. The scale bar is 250 nm.

random grain stacking. Scanning electron microscopy (SEM) analysis of the surface morphology (Fig. 2e) showed that the introduction of $K_2SnO_3$ resulted in a more uniform and densely packed large-grain structure in the seed-induced perovskite films, with significantly reduced roughness on both the upper and lower surfaces (Supplementary Fig. 9). Furthermore, energy dispersive spectrometry test results revealed that K elements were distributed throughout the entire film (Supplementary Fig. 10). Cross-sectional SEM images (Fig. 2f) revealed that the grains in the target films grew perpendicular to the substrates, exhibiting clear preferred orientation and no transverse cracks. This is due to the presence of $ABX_3$-structured $PbSnO_3$ seeds, which establish a favorable growth framework for Sn-Pb perovskite films. These seeds reduced the perovskite's interface energy, lowered the nucleation barrier, and increased the nucleation rate,

enabling perovskite nucleation at an early stage (Supplementary Fig. 11).

To study the growth kinetics of Sn-Pb perovskite films, we also performed in situ light absorption and time-tracked XRD patterns of the samples[50]. The in situ absorption spectra were measured in an $N_2$-filled glove box. The tests were conducted in two stages: (i) drop-casting an antisolvent onto spin-coated films, and (ii) annealing the perovskite wet films. As shown in Supplementary Fig. 12a, b, the film's absorption signal emerged as the antisolvent was added, with the absorption edge gradually shifting to longer wavelengths. The black arrows in Supplementary Fig. 12a, b illustrate that the target film exhibited faster changes in the rate and strength of the absorption edge migration, indicating a higher nucleation and crystallization rate[50,51]. This enhanced crystallization is attributed to the $PbSnO_3$

seeds, which reduced the nucleation barrier. During the subsequent annealing process (Supplementary Fig. 12c, d), the absorption signal of the target group reached its final state within 8 s, while the control group required over 10 s. The red dashed box in Supplementary Fig. 12c highlights a transient absorption signal (800–1000 nm) in the control film, which disappeared as annealing progressed. This phenomenon is associated with intermediate phase transitions during perovskite formation[48,52]. Time-tracked XRD measurements of the wet films (Supplementary Fig. 13) showed that the diffraction peak intensity of the intermediate phase in the control film gradually decreased, while the intensity of the perovskite diffraction peaks corresponding to the final crystal planes increased. Supplementary Fig. 13d further illustrates that, within the first 8 min, the diffraction peak intensity of the (100) crystal plane in the control group film continued to increase, and the growth rate accelerated. After 8 min, the growth rate began to slow down. For the (111) crystal plane, the diffraction peak intensity showed rapid growth after 6 min. The diffraction peak intensity of the (210) crystal plane increased only slightly. These irregular changes in diffraction peak intensity may be due to the random orientation growth of grains in the perovskite films. In contrast, the diffraction peaks of the perovskite intermediate phase were absent in the perovskite wet film of the target group. As shown in Supplementary Fig. 13c–e, the diffraction peak intensity of the (100) crystal plane continued to increase over time, while the intensity of the (111) and (210) crystal planes remained relatively stable. This indicates a clear preferential crystallization along the (100) crystal plane in the target films.

Apart from regulating crystal growth, the introduction of $K_2SnO_3$ can inhibit the oxidation of $Sn^{2+}$ in Sn-Pb perovskites. X-ray photoelectron spectroscopy (XPS) was employed to analyze the elemental composition and valence states on the upper and lower surfaces of Sn-Pb perovskites. As shown in Supplementary Fig. 14, the Sn $3d$ peak was decomposed into $Sn^{2+}$ and $Sn^{4+}$, with a significant reduction in $Sn^{4+}$ content from 23.08% (39.19%) to 7.45% (28.83%) at the top (buried) surface of the target films, effectively mitigating $Sn^{2+}$ oxidation. Moreover, the peak positions of Sn $3d$ and Pb $4f$ shifted to lower binding energies (Supplementary Fig. 15), likely due to Lewis acid-base coordination between the multi-electron donor $SnO_3^{2-}$ and $Sn^{2+}/Pb^{2+}$ element (Supplementary Fig. 3), inhibiting $Sn^{2+}$ oxidation and the formation of defect state.

In addition to incorporating $K_2SnO_3$ into perovskite precursors, it was also introduced into poly (3, 4-ethylenedioxythiophene) polystyrene sulfonate (PEDOT: PSS) solutions, which serve as hole transport materials in the devices. The strong acidic sulfonic acid groups in PEDOT: PSS can react with Sn-Pb perovskites, leading to defect formation at the buried interface of the films. These defects hinder effective carrier transport and deteriorate device stability[53,54]. Therefore, we aimed to adjust the pH of the PEDOT: PSS solution by adding $K_2SnO_3$, to alleviate its strong acidity. As shown in Supplementary Fig. 16, the pH of the aqueous solution with 3 mg mL$^{-1}$ of $K_2SnO_3$ was 4.87, nearly doubling the pH of the pure PEDOT: PSS solution (2.51). In addition, the incorporation of $K_2SnO_3$ into PEDOT: PSS improved the wettability of the perovskite precursors on PEDOT: PSS and enhanced its conductivity, which is beneficial to the transport of carriers (Supplementary Figs. 17–19 and Supplementary Table 2).

Thanks to the improved quality of the perovskite films and the interface layer, the Sn-Pb perovskite films exhibited enhanced carrier transport. As shown in Supplementary Fig. 20, for the perovskite films deposited on ITO substrates, compared to the control, the target group film exhibited stronger PL intensity and longer carrier lifetimes, regardless of whether the excitation light was incident from the top or bottom. This is due to its enhanced crystallinity and reduced non-radiative recombination defects[55] (Supplementary Table 3). However, for the perovskite films deposited on PEDOT: PSS/ITO substrates, when the excitation light was incident from the top surface, the PL

intensity of the target group increased, and the carrier lifetime rose from 1139 ns to 1371 ns. This improvement can be attributed to the following factors: (1) The introduction of $K_2SnO_3$ into PEDOT: PSS enhanced the wettability of the perovskite precursor solutions, leading to the formation of more uniform perovskite films. (2) The in situ formation of $PbSnO_3$ seeds induced preferred orientation crystallization, enhancing the crystal quality of the films. When the excitation light was incident from the PEDOT: PSS-ITO side, the target group exhibited stronger PL quenching, with the carrier lifetime decreasing from 796 ns to 700 ns. This reduction is attributed to the efficient hole carrier extraction and transport properties of PEDOT: PSS with $K_2SnO_3$ (Supplementary Table 4)[56,57].

## Analysis of the seed-induced oriented crystallization mechanisms

To understand the mechanisms behind $PbSnO_3$ seed-induced preferred orientation crystallization in Sn-Pb perovskite thin films, we performed density functional theory (DFT) calculations to determine the binding energies of $PbI_2$, $SnI_2$, and FAI on the $PbSnO_3$ (111) and $FAPb_{0.5}Sn_{0.5}I_3$ (100) surfaces, as shown in Fig. 3a–f. The binding energy of $PbI_2$ on the $PbSnO_3$ substrate was −12.08 eV (Fig. 3a), which was significantly higher than the −1.75 eV observed on the perovskite substrate (Fig. 3d), indicating a much stronger binding affinity of $PbSnO_3$ compared to the perovskite. A similar pattern was seen with $SnI_2$ and FAI, where the binding energies on the $PbSnO_3$ surface were −12.33 eV and −10.59 eV, respectively, while these values decreased to −1.69 eV and −2.19 eV on the perovskite surface.

To further explore the underlying mechanism behind the stronger surface binding of $PbSnO_3$, we calculated the planar-averaged charge density differences between substrates and perovskite precursors. For instance, $PbI_2$ on the $PbSnO_3$ substrate exhibited a strong electron-withdrawing effect, resulting in an electron accumulation of 0.27 e/Å, while $PbI_2$ itself lost 0.33 e/Å. In contrast, on the perovskite surface, the electron density accumulation was only 0.18 e/Å, with $PbI_2$ losing 0.16 e/Å. The greater charge transfer from $PbI_2$ to the $PbSnO_3$ substrate suggests stronger binding behavior, which can be attributed to the higher electronegativity of O (3.44 on the Pauling scale) compared to I (2.66 on the Pauling scale) on the respective surfaces. This O-rich surface of $PbSnO_3$ made it a more effective seed layer than perovskites.

We also compared the facet-selective adsorption effects of perovskites on the $PbSnO_3$ surface, following the method outlined in reference[58]. As shown in Fig. 3g, the perovskite (100) facet exhibited a higher negative binding energy of −0.52 eV/Å², compared to −0.24 eV/Å² for the (210) facet and −0.06 eV/Å² for the (111) facet, indicating a preferential binding to the (100) surface. This selective binding likely promotes the growth of perovskites with (100) surface termination. Our experimental results also showed enhanced XRD peak intensities for the (100) and (210) surfaces compared to that of the (111) surface (Fig. 2a, b).

## Photovoltaic characteristics of single-junction Sn-Pb PSCs

Inspired by the exceptional performance of the seed-induced oriented perovskite films, we designed p-i-n-structured (ITO/PEDOT: PSS/narrow-bandgap (NBG) perovskite/$C_{60}$/bathocuproine (BCP)/Cu) single-junction solar cell and systematically studied their photovoltaic performance. To optimize device performance, we investigated devices using various concentrations of $K_2SnO_3$ in PEDOT: PSS and perovskite precursor solutions. The optimal concentrations of $K_2SnO_3$ were determined to be 3 mg mL$^{-1}$ in PEDOT: PSS and 2 mg mL$^{-1}$ in the perovskite precursor solutions (Supplementary Figs. 21, 22). The addition of $K_2SnO_3$ into both the PEDOT: PSS and perovskite precursor solutions significantly enhanced device performance. As shown in Fig. 4a, under reverse scanning, the control device achieved an efficiency of 20.57%, while the optimized $K_2SnO_3$-incorporated device exhibited an efficiency of 23.32%, with an open-circuit voltage

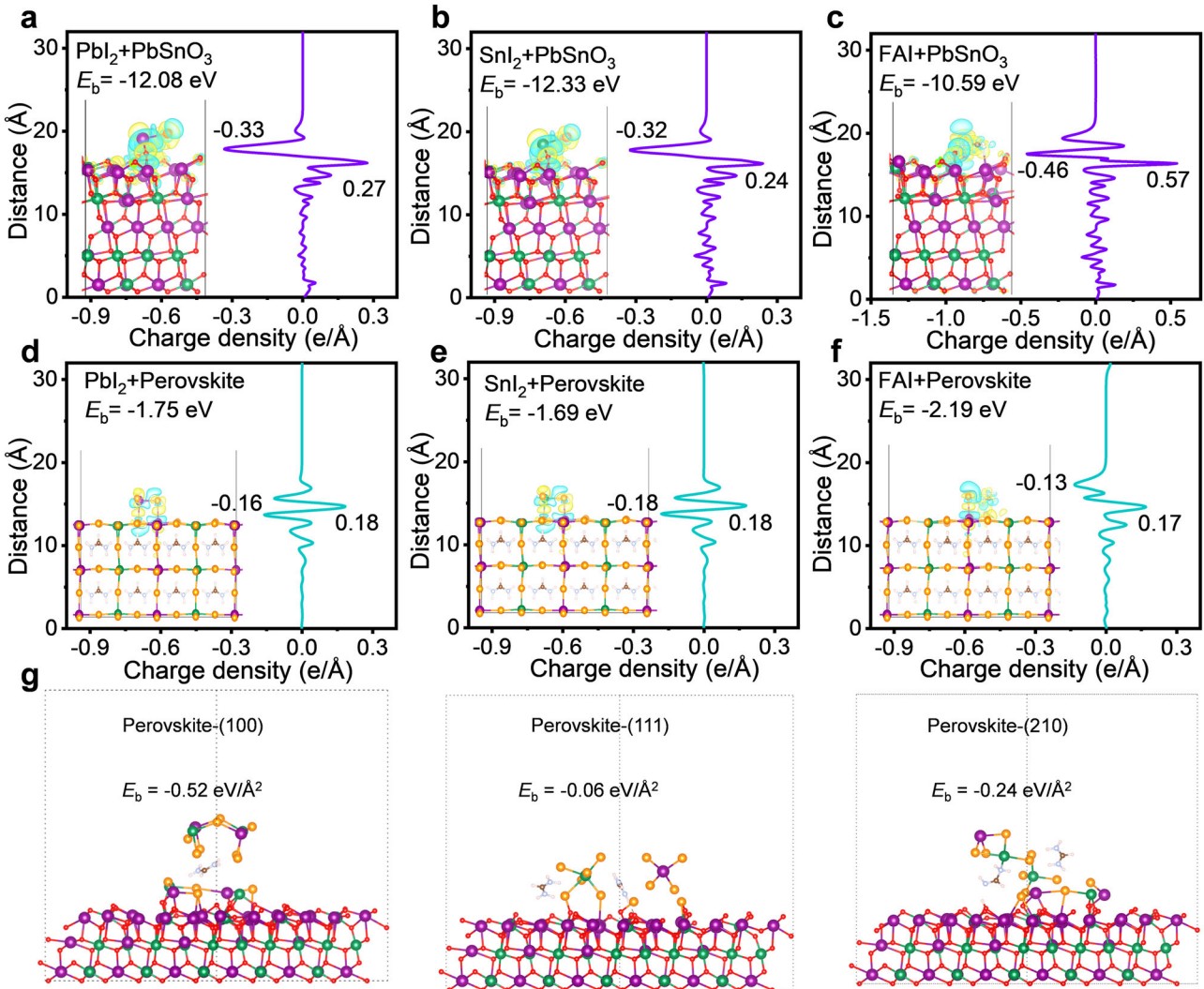

**Fig. 3 | DFT and analysis of the seed-induced oriented crystallization mechanisms. a–f** DFT-calculated binding energies ($E_b$) and planar-averaged charge density differences for (**a**) $PbI_2$, (**b**) $SnI_2$, and (**c**) FAI on the $PbSnO_3$ (111) surface; (**d**) $PbI_2$, (**e**) $SnI_2$, and (**f**) FAI on the $FAPb_{0.5}Sn_{0.5}I_3$ (100) surface. Insets in (**a–f**) show the corresponding charge density differences in real space, where positive and negative values indicate charge accumulation (yellow color) and depletion (blue color), respectively. **g** Schematic of the selective adsorption of perovskite facets on the $PbSnO_3$ (111) surface.

($V_{OC}$) of 0.88 V, a fill factor (FF) of 80.15%, and a short-circuit current density ($J_{SC}$) of 32.90 mA cm$^{-2}$. Detailed photovoltaic parameters are listed in Supplementary Table 5. External quantum efficiency (EQE) test results (Fig. 4b) showed integrated $J_{SC}$ values of 31.92 mA cm$^{-2}$ for the control group and 32.53 mA cm$^{-2}$ for the target group. The steady-state power output (SPO) efficiency at the maximum power point (MPP) for the best-performing Sn-Pb target device was 23.12% (Fig. 4c). Statistical analysis of PCE for randomly selected control and target devices revealed an increase in average PCE from 20.32 ± 0.24% to 22.92 ± 0.40% (Fig. 4d and Supplementary Fig. 23).

The synergistic effect of $K_2SnO_3$ incorporation not only improved the PCE of Sn-Pb PSCs but also significantly enhanced their stability. Long-term stability tests conducted in an $N_2$-filled glovebox revealed that the PCE of the unencapsulated control device dropped to 70% of its initial value after 1000 h of storage (Fig. 4e). In contrast, the unencapsulated target device retained 95.02% of its original PCE even after 3760 h. Moreover, MPP tracking stability tests under constant 1-sun illumination in an $N_2$-filled glovebox at ~ 55 °C demonstrated that the unencapsulated target device retained 90% of its initial efficiency after 200 h of operation (Supplementary Fig. 24), significantly outlasting the control device. These results highlight that seed-induced oriented crystallization not only improved the crystal quality of Sn-Pb

perovskite films but also effectively passivated defects, leading to both improved efficiency and long-term stability of the devices.

The significant improvement in device performance can be mainly attributed to the reduction of carrier recombination and the increase in effective transmission. As shown in Supplementary Fig. 25a, the introduction of $K_2SnO_3$ into PEDOT: PSS and perovskite precursors significantly reduced the dark current density of the devices from $4.00 \times 10^{-6}$ mA cm$^{-2}$ to $1.08 \times 10^{-6}$ mA cm$^{-2}$ at 0 V. In addition, the recombination resistance ($R_{rec}$) of the target device increased (Supplementary Fig. 25b), which contributed to suppressing carrier recombination. The built-in potential ($V_{bi}$) of PSCs was further analyzed using capacitance-voltage (C-V) measurements. The Mott-Schottky curve in Fig. 4f shows that the $V_{bi}$ of the target device (0.66 V) was higher than that of the control device (0.51 V), indicating that the addition of $K_2SnO_3$ effectively reduced the defect state density and improved carrier transport efficiency.

To further investigate carrier recombination in devices, the $V_{OC}$ versus light intensity was analyzed. As shown in the fitting results in Fig. 4g, the ideal factor for the target group (1.18) was significantly lower than that of the control group (1.45), suggesting reduced trap-assisted carrier recombination. To quantify the defect state density in perovskite films, space charge limited current measurements were

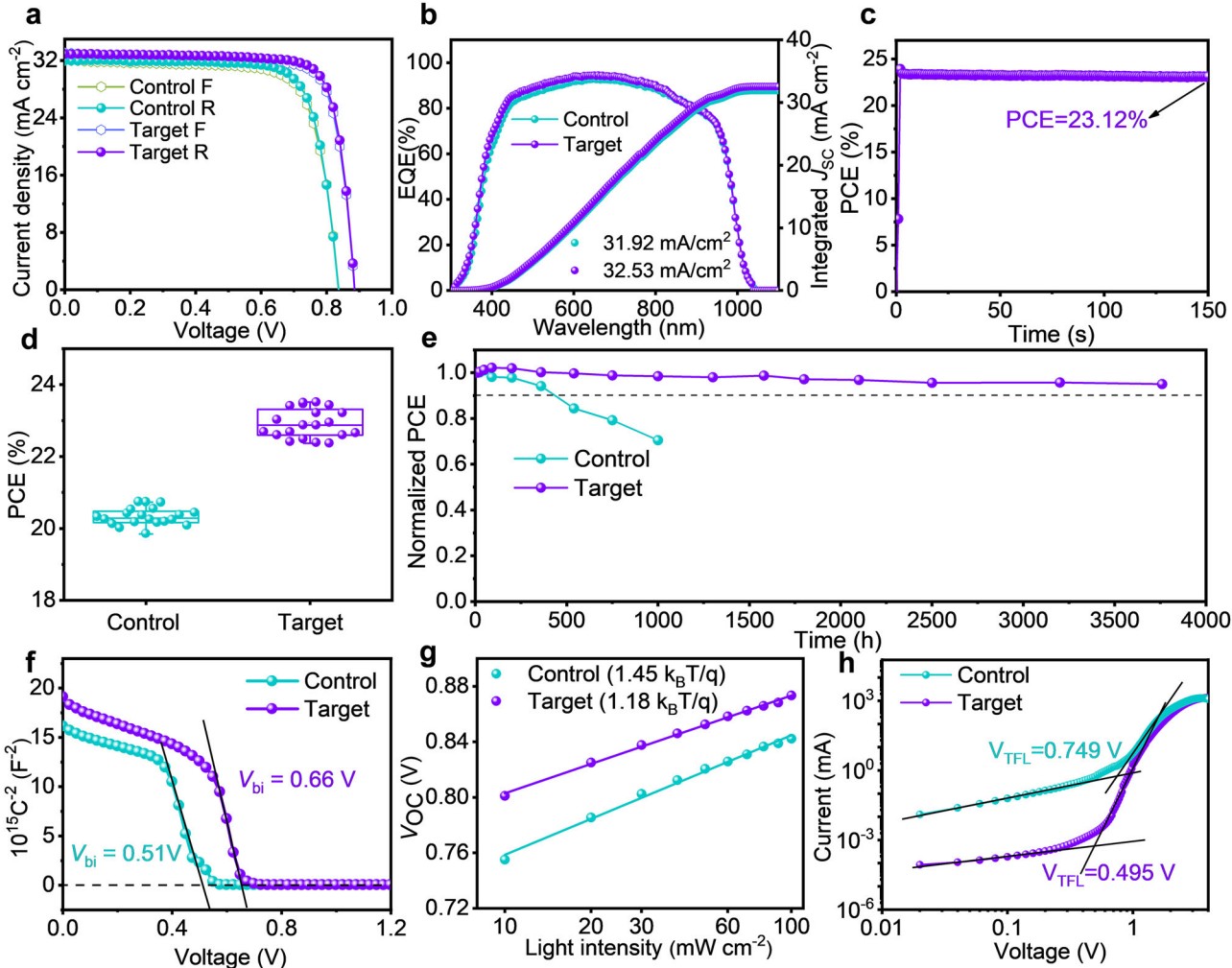

**Fig. 4 | Photovoltaic and carrier transport characteristics of single-junction Sn-Pb PSCs. a** J–V curves of control and $K_2SnO_3$-modified single-junction Sn-Pb PSCs. **b** The corresponding EQE curves of the Sn-Pb PSCs. **c** Steady-state power output (SPO) efficiency of the best-performing Sn-Pb PSC at a bias voltage of 0.76 V. **d** PCE statistics of control and target devices, with 20 devices measured for each type. **e** Long-term stability of unencapsulated control (initial efficiency: 20.24%) and target (initial efficiency: 22.63%) single-junction Sn-Pb cells stored in an $N_2$-filled glovebox. **f** Mott-Schottky curves, (**g**), ideality factor characteristics, and (**h**), J-V curves of hole-only devices without and with $K_2SnO_3$ incorporation.

conducted using hole-only devices (Glass/ITO/PEDOT: PSS/Sn-Pb perovskite/poly (3-hexylthiophene-2, 5-diyl)/Au). The defect state density ($N_{trap}$) was estimated using the following formula (2)[59].

$$V_{TFL} = \frac{qN_{trap}L^2}{2\varepsilon_r\varepsilon_0} \qquad (2)$$

Where $q$ is the elementary charge, $N_{trap}$ is the trap density, the trap-filling limit voltage ($V_{TFL}$) is the trap-filled limit voltage, $\varepsilon_r$ is the relative dielectric constant of perovskites, and $\varepsilon_0$ is the vacuum permittivity ($\varepsilon_0 \approx 8.854 \times 10^{-2}$ F m$^{-1}$), $L$ is the thickness of perovskite films. From the dark current-voltage (J-V) curves (hole-only devices) in Fig. 4h, the $V_{TFL}$ decreased from 0.749 V to 0.495 V after $K_2SnO_3$ introduction, corresponding to a decrease in $N_{trap}$ from $4.23 \times 10^{15}$ cm$^{-3}$ and $2.20 \times 10^{15}$ cm$^{-3}$. This suggests that the addition of $K_2SnO_3$, along with $PbSnO_3$ seeds, effectively passivated defects and reduced the defect concentration.

Given that the energy level distribution in perovskites plays a crucial role in carrier separation and transport, the energy band structure of the devices was analyzed using ultraviolet photoelectron spectroscopy (UPS). The UPS results demonstrated that the introduction of $K_2SnO_3$ mitigated the P-type self-doping effect caused by $Sn^{2+}$ oxidation and improved the energy level alignment in the

perovskite light-absorbing layer. Kelvin probe force microscopy (KPFM) measurements, using highly oriented graphene as a reference standard, further confirmed the reliability of the $E_f$ energy level shift observed in the UPS tests (Supplementary Figs. 26–28). In addition, both the control film and the target film exhibited similar UV-Vis-NIR absorption spectra and identical bandgaps (Supplementary Fig. 29), indicating that the composition of the Sn-Pb perovskite absorbers remained unchanged.

**Photovoltaic performance of all-perovskite tandem solar cells**
Building on the excellent performance of NBG PSCs, they were combined with WBG PSCs to form all-perovskite tandem solar cells. The device structure is illustrated in Fig. 5a. As shown in Fig. 5b and Supplementary Table 6, the best-performing tandem device exhibited a reverse (forward)-scanned PCE of 28.20% (27.85%), with a $V_{OC}$ of 2.14 V (2.14 V), a $J_{SC}$ of 15.89 mA cm$^{-2}$ (15.82 mA cm$^{-2}$), and an FF of 83.07% (82.34%). The device also demonstrated a stabilized efficiency of 28.12%, measured over 310 s (Fig. 5c). The cross-sectional SEM image of the solar cell (inset of Fig. 5c) revealed that the Sn-Pb perovskite layer had excellent crystallinity and a clear orientation. The integrated $J_{SC}$ values from EQE spectra for the top WBG subcell and bottom NBG subcell were 15.83 and 15.62 mA cm$^{-2}$, respectively (Fig. 5d), closely

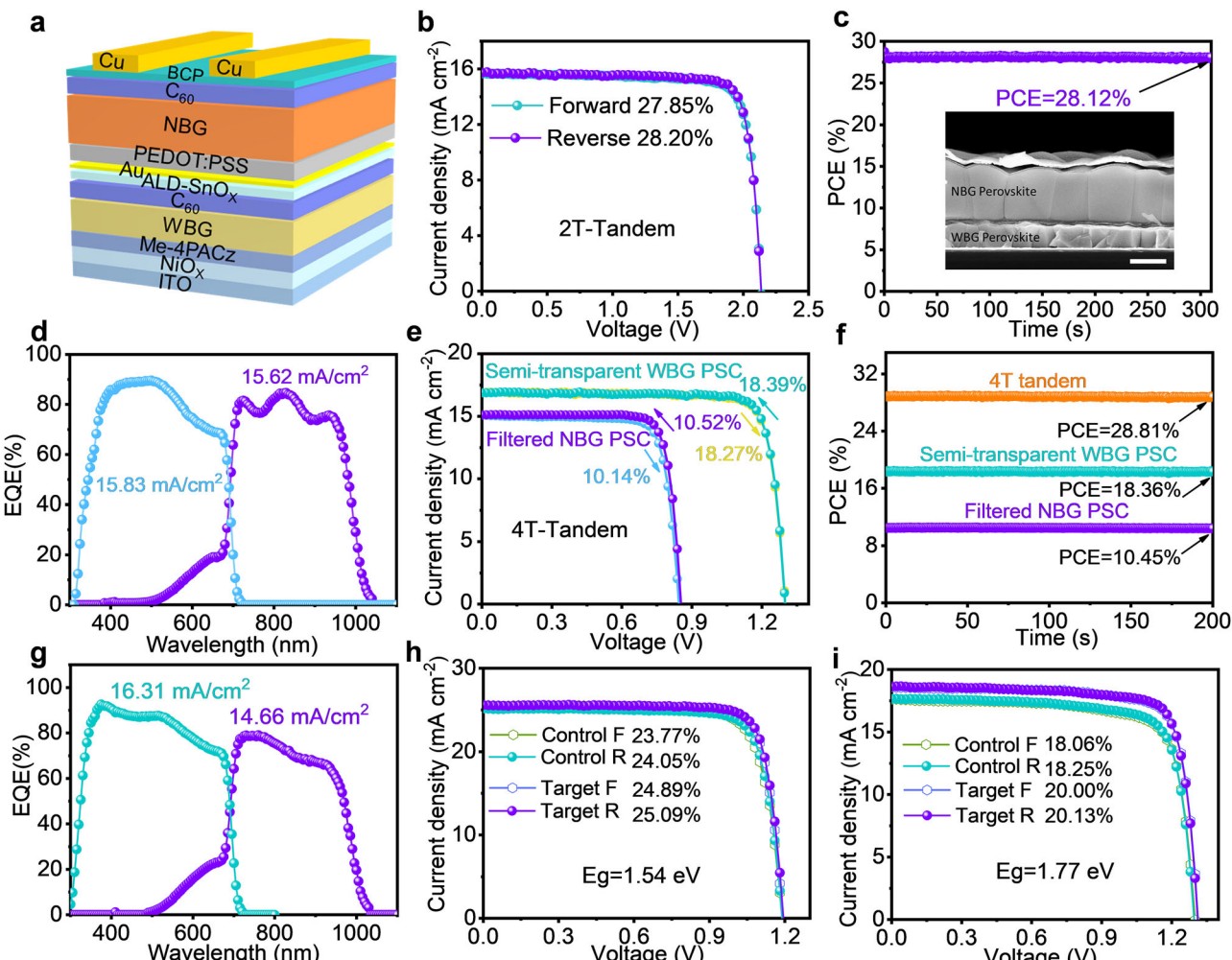

**Fig. 5 | Photovoltaic performance of all-perovskite tandem solar cells and other bandgap perovskites. a** Device architecture diagrams of all-perovskite 2 T tandem solar cells. **b** *J–V* curves of the champion 2 T tandem PSCs. **c** SPO measured at a bias voltage of 1.9 V for the best-performing 2 T tandem PSC. Illustration: Cross-sectional SEM image of a 2 T tandem solar cell (The scale bar is 600 nm). **d** EQE curves of the 2 T tandem PSCs. **e** *J–V* curves of two different devices: a single-junction semi-transparent WBG PSC, and a filtered NBG PSC. **f** SPO measured at bias voltages of 1.16 V and 0.74 V for the semi-transparent WBG PSC and the filtered NBG PSC, respectively. **g** EQE curves of the single-junction semi-transparent WBG PSC and the filtered NBG PSC. **h** *J–V* curves of control and K$_2$SnO$_3$-modified 1.54 eV-bandgap single-junction PSCs. **i** *J–V* curves of control and K$_2$SnO$_3$-modified 1.77 eV WBG single-junction PSCs.

matching the $J_{SC}$ values obtained from *J-V* measurements. To demonstrate reproducibility, 32 individual all-perovskite tandem solar cells were fabricated and statistically analyzed, yielding an average PCE of 27.87 ± 0.31% (Supplementary Fig. 30). An independent laboratory from a third-party professional certification institution certified a reverse-scanned efficiency of 27.60% for a representative device (Supplementary Fig. 31). The unencapsulated tandem solar cell also exhibited good long-term shelf-storage stability, maintaining 95.73% of its initial PCE after 4038 h in an N$_2$-filled glovebox (Supplementary Fig. 32). Moreover, MPP tracking was conducted to assess the operational stability of the unencapsulated devices in an N$_2$-filled glovebox at approximately 55 °C. As shown in Supplementary Fig. 33, the unencapsulated all-perovskite tandem solar cell maintained 80% of its initial PCE after 800 h of operation. This enhanced stability is attributed to the significant improvements in crystal quality due to seed-induced oriented crystallization, reduced defect density, and alleviated interfacial stresses.

Furthermore, 4 T all-perovskite tandem solar cells were developed by combining NBG Sn-Pb cells with semi-transparent WBG cells (Supplementary Fig. 34). The transmittance spectrum of the semi-transparent WBG cell is shown in Supplementary Fig. 35. *J-V* curves for the semi-transparent WBG PSC and the filtered NBG PSC are presented

in Fig. 5e. The semi-transparent WBG PSC realized a PCE of 18.39% (18.27%), and the filtered NBG cell obtained a champion PCE of 10.52% (10.14%) under reverse (forward) scanning (Supplementary Table 7). As a result, the mechanically stacked 4 T all-perovskite tandem solar cell achieved an SPO efficiency of 28.81% (Fig. 5f). The $J_{SC}$ values obtained from the EQE spectra for the WBG PSC and the filtered NBG PSC were 16.31 mA cm$^{-2}$ and 14.66 mA cm$^{-2}$ (Fig. 5g), respectively, in good agreement with the $J_{SC}$ values from *J-V* measurements.

## Universal of seed-induced strategy for other bandgap perovskites

In addition to NBG Sn-Pb perovskites, our ABX$_3$ seed-induced strategy demonstrates broad applicability to other perovskite materials. K$_2$SnO$_3$-derived PbSnO$_3$ seeds were introduced into both 1.54 eV normal-bandgap and 1.77 eV WBG PSCs for testing. As shown in Fig. 5h and Supplementary Table 8, the introduction of K$_2$SnO$_3$-derived PbSnO$_3$ seeds effectively improved the performance of the 1.54 eV normal-bandgap devices. The PCE of the 1.54 eV-bandgap device increased from 24.05% to 25.09% after the introduction of K$_2$SnO$_3$. Even after 300 s of steady-state SPO testing, the efficiency remained at 25.01% (Supplementary Fig. 36). This improvement can be mainly attributed to the enhanced crystallinity of the perovskite films

(Supplementary Fig. 37) and effective defect passivation. Similarly, the introduction of $K_2SnO_3$-derived $PbSnO_3$ seeds improved the performance of the 1.77 eV WBG devices. The target device achieved an efficiency of 20.13% (Fig. 5i), and its SPO efficiency stabilized at 20.06% after 300 s (Supplementary Fig. 38). XRD test results demonstrated that the $ABX_3$ seed-induced preferred orientation crystallization was applicable to WBG perovskite films. The diffraction peak intensity of the (001) and (002) crystal planes in the target film was significantly enhanced (Supplementary Fig. 39), and the preferred orientation growth of the film facilitated more effective carrier transport.

## Discussion

The strategy of seed-induced oriented crystallization has been pivotal in regulating the nucleation and crystallization processes of Sn-Pb PSCs. This method effectively addresses the issue of random crystal orientation that often arises from the rapid growth rates of Sn-Pb perovskite crystals. Mitigating this issue enabled the development of high-performance single-junction Sn-Pb perovskite and all-perovskite tandem solar cells. The in situ construction of the $ABX_3$-structured seed layer, $PbSnO_3$, provided a robust framework for the growth of Sn-Pb perovskites. This seed layer exhibited strong binding interactions and high lattice matching with perovskites, promoting preferential orientation crystallization. As a result, perovskite films demonstrated enhanced crystallinity, preferred orientation, and reduced stresses. In addition, the incorporation of multi-electron donor $SnO_3^{2-}$ and alkali metal $K^+$ ions, coordinated with $Sn^{2+}$, $Pb^{2+}$, and $I^-$, effectively passivated defects and improved carrier transport efficiency. These improvements in crystallinity and the reduction of defect state density led to single-junction Sn-Pb PSCs with a steady-state PCE of 23.12%, along with significantly improved stability. The all-perovskite tandem solar cells, utilizing these PSCs as the bottom sub-cells, achieved impressive results, with the 2 T cells reaching a maximum PCE of 28.12%, and the 4 T cells achieving a maximum PCE of 28.81%. Notably, this method is versatile and can be applied to both 1.77 eV WBG and 1.54 eV normal-bandgap PSCs. In conclusion, the $ABX_3$ seed-induced oriented crystallization strategy provides an effective approach for enhancing the performance of perovskite photovoltaic devices, paving the way for the development of high-efficiency and stable perovskite solar cells.

## Methods

### DFT calculations

DFT calculations were carried out using the Vienna ab initio simulation package[60,61]. The core−valence interaction was described using the projector-augmented wave method[62,63]. A cutoff energy of 520 eV was set for the basis functions. The generalized gradient approximation (GGA) with the Perdew−Burke−Ernzerh (PBE) functional was used for exchange correlation[64]. The Grimme's DFT-D3 scheme was employed for the inclusion of van der Waals interactions[65]. All atoms were relaxed until the Hellmann−Feynman forces on them were below 0.03 eV Å$^{-1}$. Ordered $FAPb_{0.5}Sn_{0.5}I_3$ configuration with the lowest total energy was chosen as the matrix to approximately mimic experimental perovskite compositions. The $PbSnO_3$ and perovskite surface models were chosen with respect to their strongest XRD peaks, i.e., the (111) and (100) planes, respectively. Typically, the most stable O-rich (111) surface of $PbSnO_3$ was chosen for adsorption calculations among three different terminations. A vacuum thickness of at least 15 Å was employed to prevent potential interactions between adjacent layers. A cluster adsorption model was used to investigate the facet-selective adsorption of perovskite on top of $PbSnO_3$, following the methodology in reference[58]. The Γ-only k-point mesh and dipole correction were chosen for slab calculations. The reaction energy was calculated as the difference in total energies between products and reactants. The crystal structures of solvent-related intermediate phases were taken from reference[66]. The binding energy $E_b$ was calculated using the equation: $E_b = E_{tot}(adsorption) - E_{tot}(slab) - E_{tot}(molecule)$, where $E_{tot}$

represents the total energy of the corresponding system. The visualization of crystal structures was performed using VESTA software[67], and the VASPKIT code was used for data processing[68].

### Materials

All materials were used directly without further purification. FAI, MAI, $SnI_2$, $PbI_2$, and lead (II) bromide ($PbBr_2$) were purchased from Advanced Election Technology Co., Ltd. 1,3-propane-diammonium iodide ($PDAI_2$), Ethylenediammonium diiodide ($EDAI_2$), PEDOT: PSS (4083) aqueous solutions, BCP, and $C_{60}$ were purchased from Xi'an Polymer Light Technology. Tin fluoride ($SnF_2$), lead thiocyanate ($Pb(SCN)_2$), cesium iodide (CsI), isopropanol (IPA), DMF, DMSO, and chlorobenzene (CB) were purchased from Sigma-Aldrich. $K_2SnO_3$ was obtained from Meryer (Shanghai) Biochemical Technology Co.

### NBG $FA_{0.7}MA_{0.3}Pb_{0.5}Sn_{0.5}I_3$ perovskite precursor solutions

The precursor solutions (2.1 M) were prepared by dissolving $PbI_2$, $SnI_2$, MAI, FAI, $Pb(SCN)_2$, and $K_2SnO_3$ in a mixed DMF/DMSO (v/v, 3:1) solvent. The molar ratio of FAI: MAI was 7:3, and the molar ratio of $PbI_2:SnI_2:SnF_2:Pb(SCN)_2:K_2SnO_3$ was 1:1:0.1:0.02:0-0.014. The precursors were stirred at 35 °C for 5 h before use. All the perovskite precursors were prepared in an $N_2$-filled glovebox. Finally, the solutions were filtered with 0.22 μm polytetrafluoroethylene (PTFE) membrane before use.

### WBG $FA_{0.8}Cs_{0.2}Pb(I_{0.6}Br_{0.4})_3$ perovskite precursor solutions

The precursor solutions (1.2 M) were prepared by dissolving CsI, FAI, $PbBr_2$, and $PbI_2$ in a mixed DMF/DMSO (v/v, 4:1) solvent. The molar ratio of FAI: CsI was 8:2, and the molar ratio of $PbI_2:PbBr_2$ was 6:4. In addition, a 1 mol% concentration of $Pb(SCN)_2$ was included. Among them, 1 mg mL$^{-1}$ of $K_2SnO_3$ was added to perovskite precursors. The solutions were stirred for 2 h at 60 °C and then filtered using a 0.22 μm poly (vinylidene fluoride) membrane before use.

### Preparation of HTL solutions

HTL solutions were gained by adding approximately 0–5 mg of $K_2SnO_3$ to 1 mL of PEDOT: PSS solutions. The HTL solution was stirred on a shaker for 1 h, then filtered using a 0.45 μm filter before use.

### $FA_X MA_{1-X}PbI_3$ perovskite precursor solutions

For the 1.3 M $FA_X MA_{1-X}PbI_3$-based device fabrication, $PbI_2$ solution (1.3 M) was prepared by dissolving $PbI_2$ powder in a mixed DMF/DMSO (v/v, 19:1) solvent and stirred at 70 °C for 4 h. Among them, 1 mg mL$^{-1}$ $K_2SnO_3$ was added to the $PbI_2$ solutions of the target devices. 60 mg of FAI and 14 mg of methylamine hydrochloride (MACl) were dissolved in 1 mL IPA for the preparation of organic amine solutions.

### NBG Sn-Pb perovskite solar cell fabrication

The etched ITO substrates were sequentially cleaned with dishwashing liquid, deionized water, acetone, isopropyl alcohol, and ethanol for 15 min each. Cleaning treatment was performed using a UV plasma ozone cleaning machine for 15 min. The filtered PEDOT: PSS aqueous solutions were spin-coated onto the ITO substrates at 5000 rpm for 30 s, and subsequently annealed on a hotplate at 140 °C for 20 min. After cooling, the ITO substrates were immediately transferred into an $N_2$-filled glovebox (with controlled $H_2O$ and $O_2$ levels, both < 0.01 ppm) for perovskite film fabrication. In the following, the $FA_{0.7}MA_{0.3}Pb_{0.5}Sn_{0.5}I_3$-based precursors without or with $K_2SnO_3$ were spin-coated onto the substrates at 1000 rpm for 10 s with an acceleration of 200 rpm s$^{-1}$ and then at 4000 rpm for 40 s with an acceleration of 1000 rpm s$^{-1}$. During the second step of spin-coating, 400 μL of CB was dropped onto the spinning ITO substrates at the 30$^{th}$ s. The substrates with perovskite wet films were then annealed on a hotplate at 100 °C for 10 min. In the following, post-treatment solutions ($EDAI_2$ in IPA with a concentration of 0.5 mg mL$^{-1}$) were spin-coated onto the

perovskite films at 5000 rpm for 30 s, and the films were annealed at 100 °C for 7 min. Finally, C60 (20 nm), BCP (7 nm), and Cu (100 nm) were sequentially deposited onto the perovskite films using a thermal evaporator (Wuhan PD Vacuum Technologies Co., Ltd).

## Fabrication of WBG PSCs

Cleaned ITO substrates were treated with ultraviolet ozone for 15 min before use. Subsequently, solutions of self-assembled monolayer (SAM) Me-4PACz/MeO-2PACz (0.3 mg mL$^{-1}$ dissolved in absolute ethanol) were spin-coated onto the glass/ITO substrates at 3000 rpm for 30 s, followed by annealing at 100 °C for 10 min. Next, 40 μL of WBG perovskite precursor solutions were dropped on the substrates and spin-coated in two steps: first at 1000 rpm for 10 s and then at 5000 rpm for 60 s. At the 45$^{th}$ s mark of the second step, 350 μL of diethyl ether was dripped onto the substrates. The films were then annealed at 60 °C for 2 min, followed by 100 °C for 10 min. PDAI$_2$ solutions (80 μL, 1 mg mL$^{-1}$ dissolved in IPA) were subsequently spin-coated onto the as-prepared perovskite films at 4000 rpm for 30 s, followed by annealing at 100 °C for 5 min.

To complete the solar cell fabrication, all substrates were transferred into a thermal evaporation chamber and coated with 22 nm of C$_{60}$. For semi-transparent cells, 20 nm of atomic layer deposition (ALD) SnO$_x$ was used instead of BCP (7 nm). The precursors for ALD SnO$_x$ were tetrakis(dimethylamino), tin (IV), and deionized water. 100 nm of ITO was sputtered at a power of 100 W under an Ar pressure of 2 mTorr. The active area of the devices was defined as 0.070225 cm$^2$ and was determined by the overlapping region between the back electrode and the patterned ITO substrate. Finally, 100 nm of silver (Ag) was evaporated successively at $2 \times 10^{-4}$ Pa.

## All-perovskite tandem solar cell fabrication

The ITO substrates were treated as described above before use. NiOx nanoparticle solutions (10 mg mL$^{-1}$ NiO$_x$ in pure water) were then spin-coated onto the cleaned ITO substrates at 3000 rpm for 30 s and annealed in ambient air at 130 °C for 30 min. Next, self-assembled monolayers of Me-4PACz solutions (0.3 mg mL$^{-1}$ in ethanol) were spin-coated onto the ITO/NiOx substrates at 3000 rpm for 30 s and then annealed at 100 °C for 10 min. For the fabrication of perovskite films, 50 μL of WBG perovskite precursor solutions were dropped on each ITO substrate and spin-coated in two steps: first at 2000 rpm for 10 s and then at 6000 rpm for 40 s. 350 μL of CB was dripped at 30 s before the end of the spinning process, and the substrates were heated at 60 °C for 2 min, followed by 100 °C for 10 min. After the substrates were cooled, post-treatments with PDAI$_2$ (2 mg mL$^{-1}$ in IPA) were conducted via spin-coating a solution at 4000 rpm for 30 s, followed by annealing at 100 °C for 10 min. After cooling, the substrates were transferred to an evaporation system, where a C$_{60}$ film (18 nm) was deposited onto the WBG perovskites through thermal evaporation. ALD SnO$_x$ layers with a thickness of 20 nm were then deposited on the WBG perovskite films. Then, it was transferred to the vacuum evaporation chamber, and 0.8 nm Au was evaporated. The substrates were then transferred to an N$_2$-filled glovebox for the fabrication of NBG films, which were deposited and treated as described above. Finally, C$_{60}$ (20 nm), BCP (7 nm), and Cu (100 nm) were sequentially deposited on top of the NBG perovskite films (Wuhan PDVacuum Technologies Co., Ltd).

## FA$_X$MA$_{1-X}$PbI$_3$-based device fabrication

ITO substrates were used after processing according to the previous steps. Tin (II) Oxide (SnO$_2$) colloid solutions (15 wt% SnO$_2$: deionized water: hydrogen peroxide in a 1:4:1 v:v ratio) were spin-coated onto the substrates at 4000 rpm for 30 s and then annealed at 180 °C for 30 min. After cooling to room temperature, the substrates were transferred to an N$_2$-filled glovebox, where PbI$_2$ solutions were spin-coated onto the SnO$_2$ substrates at 1500 rpm for 30 s and annealed at 70 °C for 1 min.

Subsequently, organic amine salt solutions were spin-coated onto the PbI$_2$ films at 1500 rpm for 30 s and annealed in air (30-40 RH%, 145 °C, 13 min). For n-butylammonium bromide (BABr) post-treatment, BABr solution (3 mg mL$^{-1}$ in IPA) was spin-coated at 3000 rpm for 30 s, followed by annealing at 100 °C for 1 min. Finally, hole transport layers were prepared by spin-coating 2,2,7,7-tetrakis (N, N-di-p-methoxyphenylamine)-9,9-spirobifluorene solutions at 3000 rpm for 20 s. A 70 nm layer of Au was then sequentially deposited on top of the HTLs at a pressure of $2 \times 10^{-4}$ Pa.

## Film characterization

Crystal characteristics of perovskites and related chemical products were analyzed using an XRD instrument (Bruker AXS, D8 ADVANCE). Characterizations included XRD, pole figure, and GIXRD measurements. DLS spectra of precursors were tested by a nanoparticle analyzer (Zetasizer Nano ZSP). The morphology of the perovskite films was observed by a SEM (Tescan AMBER). FTIR spectra were tested by an FTIR spectrometer (NICOLET iS50 FTIR). In situ UV-Vis-NIR absorption spectrum measurements were conducted using a self-built test system, which included a tungsten halogen lamp, a lens, a fiber-optic setup, a spectrometer (Ocean Optics, USB2000 + fiber spectrometer), and a computer. The excitation light source was delivered via an optical fiber, emitting light at a fixed position on the substrates. The light passing through the sample was collected by the lens, transmitted to the spectrometer through the optical fiber, and subsequently processed by the computer. During the spin coating and annealing processes, the integration time of the spectrometer was set to 500 ms, with an acquisition frequency of 2 times s$^{-1}$. XPS tests were conducted with a photoelectron spectrometer (Thermo Scientific, ESCLAB 250Xi). XPS spectra were fitted using the Thermo Avantage software. UPS was measured using the same system with a He I UV source. Atomic force microscopy (AFM) height images, KPFM potential distribution images, and conducting atomic force microscopy (CAFM) measurements were obtained using an atomic force microscope (Bruker Dimension ICON XR AFM). Steady-state PL measurements were conducted with a carrier spectrometer (Horiba iHR320). Absorption spectra were collected by a UV-Vis-NIR spectrophotometer (SHIMADZU, mini1280).

## Device characterization

*J-V* curves and steady-state power outputs of PSCs were collected using a Keithley 2400 source meter under AM 1.5 G illumination (100 mW cm$^{-2}$) produced by a solar simulator (Enlitech, SS-X50). The solar simulator was calibrated to a 100 mW cm$^{-2}$ light intensity by a silicon reference solar cell (SRC-2020, Enlitech; traceable to NREL). All devices were measured for *J−V* curves with an active area of 0.0948 cm$^2$ masked using a metal mask (0.070225 cm$^2$) in an N$_2$-filled glovebox. The *J-V* measurements of PSCs were performed with a scanning rate of 0.02 V s$^{-1}$, a voltage step of 20 mV, and a delay time of 25 ms. Electrochemical impedance spectroscopy (EIS) and Mott−Schottky plots were given by a CHI 770E electrochemical workstation (Shanghai Chenhua Instruments, China). EQE data was collected using a QE/ Incident photon-to-electron conversion efficiency (IPCE) system (Enli Technology Co. Ltd) calibrated with a standard silicon cell.

## Reporting summary

Further information on research design is available in the Nature Portfolio Reporting Summary linked to this article.

# Data availability

The main data supporting the findings of this study are available within the published article and its Supplementary Information. All other data are available from the corresponding authors on request. Source data are provided in this paper.

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

## Acknowledgements

This work was supported by the Key Lab of Artificial Micro-and Nano-Structures of the Ministry of Education of China, Wuhan University. The authors thank the Core Facility of Wuhan University for PL, FTIR, AFM, SEM, DLS, and XPS measurements. The authors thank Dr. Ying Zhang from the Core Facility of Wuhan University for her help with SEM measurements and morphology analysis. The authors also acknowledged the financial support from the National Natural Science Foundation of China (No. 12174290, W.K., No. 210972127, S.Z.), Knowledge Innovation Program of Wuhan-Shuguang Project (Grant Numbers: 2023010201020245, W.K.).

## Author contributions

W.C. and W.K. conceived and directed the overall project. W.C. fabricated NBG PSCs and all-perovskite tandem cells and performed film and device characterizations. S.Z. assisted in characterizing device performance and article frame analysis. H.C. helped with the fabrication of WBG subcells in all-perovskite tandem cells. W.M. carried out the DFT analysis. D.P. assisted in SEM, XPS, and EIS measurements. H.G., G.Z., Y.G., S.C., Z.Y. (Zixi Yu), L.H., J.Z., G.C., G.L., H.F., Z.Y. (Zhiqiu Yu), and H.Z. offered help in device fabrication and film characterizations. W.C., W.M., and W.K. wrote the manuscript. All authors discussed the results and contributed to the revisions of the manuscript. G.F. and W.K. supervised this study.

## Competing interests

The authors declare no competing interests.
