## [Transparent Peer Review file · Nature Communications]

Universal In-situ Oxide-based ABX₃-structured Seeds for Templating Halide Perovskite Growth in All-perovskite Tandems

Corresponding Author: Professor Weijun Ke

Version 0:

Reviewer comments:

Reviewer #1

(Remarks to the Author)

In this work, the authors reported an interesting work - utilized K₂SnO₃ as an additive to promote perovskite growth, realizing single-junction Sn-Pb PSCs, 2T and 4T all-perovskite tandem solar cells with the PCE of 23.12%, 28.12% and 28.81%, respectively. Although the work is interesting, and the efficiencies of devices are high, the paper is poorly written and the characterizations are incomplete. For the standard of Nature Communications, the authors need to make significant revision and resubmit for consideration. Below are more specific comments for the authors.

1, on Page 4, it is noticed the beginning of the Part 1 is largely repetitive with the introduction. section The authors should balance and integrate the two professionally.

2, Figure 1a shows the schematic diagram of the seed-induced oriented crystallization. The diagram shows that PbSnO₃ tend to concentrate at the bottom of the perovskite during crystallization process. How do the authors get such conclusion? This is important for elucidating the mechanism of this work, please prove.

3, The XRD patterns of the Pbl₂ and perovskite precursors without and with K₂SnO₃ are shown in Figure 1b. The XRD peaks of the Pbl₂ and perovskite should also be included for comparison.

4, Figure 1d shows the introduction of K₂SnO₃ resulted in a significant increase in colloidal size. The authors attributed that to the formation of larger atomic clusters in the precursor solutions. What formed in this process? They also should analyse the XRD data (FWHM) in Figure 2a to confirm.

5, The authors claimed that the target films eliminated the need for intermediate-phase processes from XRD analysis. The in-situ absorption or in situ GIWAXS should be conducted to prove them. The in situ GIWAXS is also very critical to elucidate the film growth kinetics and mode of growth in the vertical direction. Angular dependent GIWAXS should be given for understand the vertical direction crystal structure of the film.

6, The authors claimed that "The strong acidic sulfonic acid groups in PEDOT: PSS can react with Sn-Pb perovskites, leading to defects formation at the buried interface of the films". Please prove them. How does the acidity of PEDOT:PSS affect the solar cell stability? Typically it is not good for stability. Please provide evidence.

7, The authors claimed that "The simultaneous introduction of K₂SnO₃ into PEDOT: PSS not only enhances the wettability of the hole transport layers but also improves its conductivity, facilitating effective carrier transport in the subsequent devices". Contact angle measurement is necessary. Please verify improved conductivity and facilitated carrier transport.

8, The authors think that the decreased PL and TRPL lifetime can be attributed to reduced non-radiative recombination. However, this is contradictory to many previous reported. The authors should measure and compare the PL and TRPL of perovskite films without and with the interface layer, respectively. Non-radiative recombination and carrier transport should be separately discussed. The unreasonable description should be deleted.

As one of the highlights is the device performance, the efficiencies of devices should be certified / third party verified.

Reviewer #2

(Remarks to the Author)

In this work, Ke et al. introduce a universal strategy for templating halide perovskite growth using oxide-based ABX₃-structured seeds, demonstrating broad applicability across various perovskite compounds. While previous studies have explored seed materials, this work presents a unique approach by incorporating potassium stannate into the perovskite precursors to form PbSnO₃ as the seed material. These PbSnO₃ seeds provide notable advantages, including strong solvent stability and high lattice compatibility with perovskites. As a result, perovskite films templated with these seeds achieve high efficiency in both single-junction and tandem solar cells. This study presents intriguing findings of broad relevance to the community, and the results are well-articulated. I recommend its publication in Nature Communications after addressing the following minor revisions.

- (1) Given that potassium stannate is introduced during perovskite film preparation, could the authors clarify the final distribution of potassium within the film? Please provide experimental evidence, such as EDS mapping, to illustrate potassium distribution in the perovskite layer.
- (2) The introduction of potassium stannate significantly enhances device performance, particularly the Voc. Could the authors provide insight into the underlying mechanism that contributes to this Voc improvement?
- (3) The reaction between potassium stannate and lead iodide yields lead stannate and potassium iodide as a by-product. Could potassium iodide also contribute positively to the film? Please discuss its potential influence within the perovskite films.
- (4) In Figure 2e, SEM results reveal residual crystals at grain boundaries in the control group. Could the authors identify the composition of these residues and explain their formation? Additionally, some voids appear at the bottom of the target perovskite films. Could the authors discuss their origin?
- (5) Potassium stannate was also introduced into the PEDOT hole transport layer to improve carrier transport. Could the authors elaborate on the rationale for this choice?
- (6) In Figure 5c, please include the scale bar value in the figure legend for clarity.
- (7) It is noted that potassium stannate addition to the perovskite precursor leads to PbSnO₃ formation. Did the precursor solution become cloudy due to the insolubility of PbSnO₃? Additionally, could the authors comment on whether adding PbSnO₃ directly would yield similar results?

Version 1:

Reviewer comments:

Reviewer #1

(Remarks to the Author)

The authors have added carefully conducted new experimental results to reply the reviewers comments. The response to questions has been done satisfactorily. I thus suggest the acceptance of the work.

Reviewer #2

(Remarks to the Author)

The author has made the necessary revisions in accordance with the feedback provided. The changes have addressed the concerns and have improved the overall manuscript. The revised version can be acceptable.

Responses to Reviewers

We would like to sincerely express our gratitude to the reviewers for dedicating their time and effort to reviewing our manuscript. The manuscript has undergone a meticulous revision, taking into account the invaluable comments provided. We have implemented the suggestions and responded to the comments in our revised manuscript. Furthermore, we have introduced supplementary experimental results and provided further explanations to bolster the persuasiveness of our conclusions.

A detailed point-by-point response to the reviewers' feedback is given below. All modifications have been marked in red font within both the revised manuscript and the supporting information.

Reviewers' Comments to Authors:

Reviewer #1:

In this work, the authors reported an interesting work - utilized K_2SnO_3 as an additive to promote perovskite growth, realizing single-junction Sn-Pb PSCs, 2T and 4T all-perovskite tandem solar cells with the PCE of 23.12%, 28.12% and 28.81%, respectively. Although the work is interesting, and the efficiencies of devices are high, the paper is poorly written and the characterizations are incomplete. For the standard of Nature Communications, the authors need to make significant revision and resubmit for consideration. Below are more specific comments for the authors.

Response: We sincerely thank the reviewer for taking the time to conduct a thorough review of our manuscript. We greatly appreciate the positive feedback and are grateful for the invaluable suggestions, which have significantly contributed to improving the quality of our work. In response to the reviewer's general comment, we have made substantial revisions to the manuscript to improve its clarity and readability.

1, on Page 4, it is noticed the beginning of the Part 1 is largely repetitive with the introduction section. The authors should balance and integrate the two professionally.

Response: Thank you for pointing out this. We acknowledge this redundancy and have revised the manuscript to eliminate overlap. Specifically, we have streamlined the content in Part 1 to avoid repeating information already covered in the introduction. The revised Part 1 now provides a focused discussion that builds upon the introduction, ensuring better integration and a smoother transition. We believe these changes improve the overall flow and professionalism of the manuscript.

Kindly refer the revised sentences on Page 5 of our revised manuscript.

2, Figure 1a shows the schematic diagram of the seed-induced oriented crystallization. The diagram shows that $PbSnO_3$ tend to concentrate at the bottom of the perovskite during crystallization process. How do the authors get such conclusion? This is important for elucidating the mechanism of this work, please prove.

Response: We thank the reviewer for raising this important question. $PbSnO_3$ has very limited solubility in perovskite precursor solvents, as shown in Figure R1. Most of the

PbSnO₃ added to the DMF/DMSO mixed solvents precipitates to the bottom of the container, likely due to gravitational effects. Consequently, we hypothesize that PbSnO₃ predominantly localizes at the bottom of the films.

To confirm this hypothesis, we conducted grazing incidence X-ray diffraction measurements to analyze the angular diffraction patterns of perovskite films treated with K₂SnO₃. As the incidence angle (ω) gradually increased, the detection depth also deepened. When $\omega=1.5^\circ$, the diffraction peak of PbSnO₃ was observed. As ω further increased further to 2° , the diffraction peak of the ITO substrate became evident, indicating that the ITO glass substrate was being detected. At this point, the diffraction signal of PbSnO₃ was further enhanced, and the ratio of the integrated peak areas of the PbSnO₃ to the (100) perovskite increased from 0.13:1 to 0.18:1. These results confirm that PbSnO₃ was more concentrated near the bottom interface of the final film. These findings are included in Figure S6 of the revised Supporting Information.

Figure R1. Photograph illustrating PbSnO₃ dispersed in DMF and DMSO solvents.

Supplementary Figure S6. Grazing incident X-ray diffraction (GIXRD) patterns of the top surface of a K₂SnO₃-incorporated perovskite film.

3, The XRD patterns of the PbI₂ and perovskite precursors without and with K₂SnO₃ are shown in Figure 1b. The XRD peaks of the PbI₂ and perovskite should also be included for comparison.

Response: We appreciate the reviewer's valuable suggestion. In response, we have added the XRD patterns of PbI₂ and perovskites to facilitate a clearer and more comprehensive comparison.

Figure 1b. XRD patterns of PbI₂, the products following the reaction of K₂SnO₃ with PbI₂, and perovskites treated without and with K₂SnO₃ (PbSnO₃: PDF 17-0607, KI: PDF 04-0471).

4, Figure 1d shows the introduction of K₂SnO₃ resulted in a significant increase in colloidal size. The authors attributed that to the formation of larger atomic clusters in the precursor solutions. What formed in this process? They also should analyse the XRD data (FWHM) in Figure 2a to confirm.

Response: We thank the reviewer for the valuable and perceptive suggestions. We hypothesize that adding K₂SnO₃ to the perovskite precursor's results in the formation of KI and PbSnO₃. Once PbSnO₃ is formed, the perovskite precursor salts preferentially adsorb onto the PbSnO₃ surface, facilitating the formation of larger clusters (Figure 3).

To validate this hypothesis, we varied the concentration of K₂SnO₃ in the precursor solution. As the amount of K₂SnO₃ increased, the solution gradually became cloudy, and the Tyndall effect became more pronounced, indicating an increase in aggregate density within the precursor solution (Figure R2a-b) (*Nat. Synth.*, 2024, <https://doi.org/10.1038/s44160-024-00687-2>, *Angew. Chem. Int. Ed.*, 2024, 63(35): e202408586.). Moreover, when the PbSnO₃ concentration reached 5 mg mL⁻¹, sediments were observed at the bottom of the glass bottle. XRD analysis of the sediment (Figure R2c) confirmed the presence of K₂SnO₃, PbSnO₃, and KI. These findings are consistent with our results shown in Figure 1b and align with the reaction mechanism outlined in Figure 1c and Equation (1):

Additionally, DFT calculations (Figure 3) revealed that perovskite precursors tend to bind to PbSnO₃, further supporting our hypothesis. These results collectively indicate that PbSnO₃ formed in the precursor solutions, along with the perovskite precursors preferentially adsorbing onto its surface, facilitates the formation of atomic clusters.

We also analyzed the diffraction peak positions in Figure 2a and found no shifts

after adding K_2SnO_3 , confirming that K_2SnO_3 does not participate in the lattice composition of the perovskite. Furthermore, comparing the FWHM of the diffraction peaks was compared (Figure S4) showed that the target group exhibited smaller FWHM compared to the control group, indicative of improved crystallinity. Notably, the (100) and (200) crystal planes demonstrated enhanced orientation, attributed to the role of PbSnO_3 as a seed material in inducing preferred orientation crystallization (Figure 3g) of the perovskite and improving the crystal quality of the films.

Figure R2. Photographs of 0 mg mL^{-1} (C), 0.5 mg mL^{-1} (T₁), 2 mg mL^{-1} (T₂) and 5 mg mL^{-1} (T₃) K_2SnO_3 added to perovskite precursor solutions under (a) natural light and (b) laser irradiation. The photos were taken after standing for 1h. (c) XRD patterns of K_2SnO_3 powder and sediments at the bottom of the T₃ bottle in figure (a).

Supplementary Figure S4. Full width at half maximum (FWHM) of the corresponding diffraction peaks of perovskite thin films from Figure 2a.

5, The authors claimed that the target films eliminated the need for intermediate-phase processes from XRD analysis. The in-situ absorption or in situ GIWAXS should be conducted to prove them. The in situ GIWAXS is also very critical to elucidate the film growth kinetics and mode of growth in the vertical direction. Angular dependent GIWAXS should be given for understand the vertical direction crystal structure of the film.

Response: We are grateful for the valuable suggestions.

At present, we are unable to perform in-situ GIWAXS measurements due to the limited availability of testing facilities. However, to study the growth kinetics in greater detail, we have developed a custom system for in-situ absorption and XRD measurements (*Nat. Commun.*, 2025, 16, 190. <https://doi.org/10.1038/s41467-024-55414-4>).

The in-situ absorption spectra were measured within an N₂-filled glovebox. The test was conducted in two stages: (i) drop-casting an antisolvent onto spin-coated films, and (ii) annealing the perovskite wet films. As shown in Figure S12a-b, the film's absorption signal emerged as the antisolvent was added, with the absorption edge gradually shifting to longer wavelengths. The black arrows in Figure S12a-b illustrate that the target film exhibited faster changes in the rate and strength of the absorption edge migration, indicating a higher nucleation and crystallization rate. These results are consistent with the in-situ microscopy results shown in Figure S11. This enhanced crystallization is attributed to the PbSnO₃ seeds, which reduce the nucleation barrier. During the subsequent spin coating process, as the solvent evaporated, the absorption intensity evolved. Before spin-coating concluded, a reduction in absorption intensity between 850-1000 nm was observed in the control film, likely due to rapid aggregation and partial re-dissolution of perovskite crystals (*Adv. Mater.*, 2024: 2411677. *Nano Lett.*, 2022, 22(18): 7545-7553.). In contrast, the target group exhibited less attenuation in absorption intensity, indicating improved initial perovskite crystal stability and film uniformity.

During the annealing stage (Figure S12c-d), the absorption signal of the target group reached its final state within 8 s, while the control group required over 10 s. The red dashed box in Figure S12c highlights a transient absorption signal (800-1000 nm) in the control film, which disappeared as annealing progressed. This phenomenon is associated with intermediate phase transitions during perovskite formation (*Nano-Micro Lett.*, 2022, 14(1): 165. *Nat. Mater.*, 2014, 13(9): 897-903. *Adv. Mater.*, 2024, 36(17): 2307635.). In contrast, the target group film did not exhibit prominent absorption signals related to intermediate phases (Figure S12d). Instead, it demonstrates stable absorption corresponding to the bandgap, maintaining consistent intensity during annealing. This suggests that the PbSnO₃ seeds eliminated intermediate phase processes, enabling stable, preferred orientation crystallization (Figure S5).

Supplementary Figure S12. In-situ light absorption evolution of Sn-Pb perovskite films, without and with K_2SnO_3 modification, during (a) (b) spin-coating and (c) (d) annealing stages.

Unfortunately, the synchrotron radiation light source test institutions are currently very busy, with new appointments scheduled for much later. Additionally, Sn-Pb perovskite precursors are prone to oxidation and instability, which makes performing in-situ GIWAXS testing extremely challenging. However, we used a specially designed water-oxygen-isolated sample box to conduct time-tracked XRD measurements, enabling us to study the nucleation and growth kinetics of perovskite films. The films were exposed to room temperature after antisolvent deposition, without annealing, and the changes in XRD patterns were monitored over time.

As shown in Figure S13b, over time, the diffraction peak intensity of the intermediate phase in the control film gradually decreased, while the intensity of the perovskite diffraction peaks corresponding to the final crystal planes increased. Figure S13d further illustrates that, within the first 8 min, the diffraction peak intensity of the (100) crystal plane in the control group film continued to increase, and the growth rate accelerated. After 8 min, the growth rate began to slow down. For the (111) crystal plane, the diffraction peak intensity initially increased, then decreased during the first 6 minutes, and showed rapid growth after 6 min. The diffraction peak intensity of the (210) crystal plane increased only slightly. These irregular changes in diffraction peak intensity may be due to the random orientation growth of grains in the perovskite films.

In contrast, the diffraction peaks of the perovskite intermediate phase were absent in the perovskite wet film of the target group. As shown in Figure S13c-e, the diffraction peak intensity of the (100) and (200) crystal planes continued to increase over time, while the intensity of the (111) and (210) crystal planes remained relatively stable. This

indicates a clear preferential crystallization along the (100) crystal plane. This behavior is attributed to the selective adsorption of perovskite on the PbSnO_3 surface. As shown in Figure 3g, the binding energy of the perovskite (100) surface is $-0.52 \text{ eV}/\text{\AA}^2$, while the binding energies of the (210) and (111) surfaces are $-0.24 \text{ eV}/\text{\AA}^2$ and $-0.06 \text{ eV}/\text{\AA}^2$, respectively. The stronger binding energy of the (100) surface likely promotes the growth of perovskite films with a (100) surface termination. As a result, Sn-Pb perovskite films with a (100) preferred orientation were successfully obtained.

Supplementary Figure S13. (a) Photograph of the sample setup for time-tracked XRD testing. The samples were placed in a sealed box to prevent exposure to water and oxygen in the air. XRD patterns of Sn-Pb perovskite wet films prepared (b) without and (c) with K_2SnO_3 treatment, showing time evolution. (♣: perovskite intermediate phase) (d) and (e) Diffraction peak intensity versus time for different crystal planes in (b) and (c), respectively.

6, The authors claimed that “The strong acidic sulfonic acid groups in PEDOT: PSS can react with Sn-Pb perovskites, leading to defects formation at the buried interface of the films”. Please prove them. How does the acidity of PEDOT: PSS affect the solar cell stability? Typically it is not good for stability. Please provide evidence.

Response: We thank the reviewer for the valuable and insightful suggestions.

During long-term testing, PEDOT: PSS with sulfonic acid groups ($-\text{SO}_3\text{H}$) can lead to the degradation of halide perovskites due to its hygroscopicity and strong acidity. This degradation results in the formation of iodine vacancies (V_I), tin vacancies (V_Sn) and lead vacancies (V_Pb), which increase non-radiative recombination, hinder carrier extraction, and affect device stability (*Joule*, 2024, 8, 2220-2237. *Sci. Adv.*, 2024, 10(16): ead12063. *Nano Energy*, 2024: 109664.). To verify these observations, we

designed a series of experiments comparing the stability of perovskite films on PEDOT: PSS films with and without K_2SnO_3 by measuring PL, XRD, and light absorption under different aging conditions.

When K_2SnO_3 is dissolved in water, it generates hydroxide ions, resulting in an alkaline aqueous solution. Therefore, we aimed to adjust the pH of the PEDOT: PSS solution by adding K_2SnO_3 , to alleviate its strong acidity. As shown in Figure S16, the pH of the aqueous solution with 3 mg mL^{-1} of K_2SnO_3 was 4.87, nearly doubling the pH of the pure PEDOT: PSS solution (2.51).

Figure R3 presents the PL emission evolution of perovskite films deposited on different HTLs. As shown, the PL intensity of the control group film decreased significantly with prolonged testing, with nearly a 40% attenuation after 120 min. However, the PL intensity of the perovskite film deposited on K_2SnO_3 -incorporated PEDOT: PSS was only attenuated by 5%, indicating a significant improvement in film stability. We infer that the rapid decay in the control group may be linked to the strong acidity of PEDOT: PSS, which can trigger iodide redox reactions ($2\text{I}^- \rightarrow \text{I}_2$) and tin oxidation reactions ($\text{Sn}^{2+} \rightarrow \text{Sn}^{4+}$) at the buried interface of Sn-Pb perovskites. These redox reactions produce a large number of vacancy defects (*Joule*, 2024, 8, 2220-2237. *Nano Energy*, 2024: 109664.).

To further confirm the enhanced stability, we measured XRD patterns of the films under continuous light exposure. Figure R4 shows that after continuous irradiation for 220 hours, the main phase diffraction peak of the perovskite film in the control group weakened, some diffraction peaks disappeared, and prominent PbI_2 diffraction peaks appeared. In contrast, the target group film showed only a weak PbI_2 diffraction peak, with the main phase diffraction peak remaining less affected.

To determine whether chemical degradation occurred during this process, we measured the UV-Vis absorption spectra of toluene solutions soaked with perovskite films deposited on different HTLs under continuous light exposure. Figure R5 shows that after 4 days, the absorption spectrum of the control group exhibited clear absorption peaks associated with I_2 , PbI_2 , and SnI_4 , while no such peaks appeared in the target group. This indicates that the sulfonate group in PEDOT: PSS may undergo a redox reaction with the buried bottom interface, leading to perovskite decomposition.

Supplementary Figure S16. pH values of (a) a PEDOT: PSS solution and (b) a PEDOT: PSS solution with K_2SnO_3 (3 mg mL^{-1}).

Figure R3. (a) Schematic diagram of steady-state PL measurements with excitation from the bottom of the films. Steady-state PL of perovskite films deposited on (b) untreated and (c) K_2SnO_3 -treated PEDOT: PSS/ITO substrates. The perovskite films were top-capped with polymethyl methacrylate (PMMA); During the test waiting period, the excitation light source was turned off. (d) Statistical plots of PL peak intensity from figures (b) and (c).

Figure R4. (a) XRD patterns of perovskite films deposited on untreated and K_2SnO_3 -treated PEDOT: PSS/ITO substrates before and after 220 h of continuous illumination. The incident light was continuously irradiated from the PEDOT: PSS/ITO side of the perovskite films.

Figure R5. UV-vis absorption spectra of toluene solutions soaked with Sn-Pb perovskite films on PEDOT: PSS/ITO substrates after aging for 4 days.

7, The authors claimed that “The simultaneous introduction of K_2SnO_3 into PEDOT: PSS not only enhances the wettability of the hole transport layers but also improves its conductivity, facilitating effective carrier transport in the subsequent devices”. Contact angle measurement is necessary. Please verify improved conductivity and facilitated carrier transport.

Response: We appreciate the valuable suggestions provided by the reviewer.

In response, we have included contact angle measurement results in Figure S17. It can be observed that after incorporating K_2SnO_3 , the contact angles of the perovskite precursors on the HTL-coated substrates decreased, which facilitates the fabrication of more uniform perovskite films.

We also performed electrical measurements on the HTL films. As shown in Figure S18a-b, the conductive atomic force microscopy (CAFM) results indicated that the introduction of K_2SnO_3 effectively enhanced the conductivity of the PEDOT: PSS film, increasing from 481 pA in the control group to 728 pA. Additionally, based on the J - V test results, the conductivity of the PEDOT: PSS film increased from 5.4×10^{-4} to $2.65 \times 10^{-3} \text{ S cm}^{-1}$, as calculated using Formula (1) (Figure S18c).

$$I = \sigma_0(A/d)V \quad (1)$$

Where A , d , and σ_0 are the sample area (0.070225 cm^2), thickness (30 nm), and electrical conductivity, respectively.

We also performed PL and TRPL measurements to confirm the improved carrier transport. As shown in Figure S19, when excitation light was incident from the upper surface of the perovskite film, the target group exhibited stronger PL intensity and longer carrier lifetime. The introduction of K_2SnO_3 improved the wettability of the perovskite precursors on PEDOT: PSS, enhancing perovskite film quality and reducing non-radiative recombination. When excited from the bottom ITO glass side, the target film showed relatively weaker PL intensity and shorter TRPL carrier lifetime due to more efficient hole extraction (Figure S19d, Table S2). This improvement is attributed to the incorporation of K_2SnO_3 , which enhances the conductivity of PEDOT: PSS and facilitates carrier transport at the buried interface.

Supplementary Figure S17. Contact angle measurements of Sn-Pb perovskite precursor solutions on ITO/PEDOT: PSS substrates. (Control: PEDOT: PSS, Target: PEDOT: PSS+3 mg mL⁻¹ K_2SnO_3).

Supplementary Figure S18. Conductive AFM images of PEDOT: PSS (a) without and (b) with K_2SnO_3 modification. (c) Electrical conductivity measurements of PEDOT: PSS and PEDOT: PSS+ $3\text{ mg mL}^{-1} K_2SnO_3$ HTLs with a structure of ITO/HTL/Cu.

Supplementary Figure S19. Schematic diagram of PL measurements with excitation from the (a) top and (b) bottom of the films. (c) Steady-state and (d) time-resolved PL spectra of pristine Sn-Pb perovskite films on PEDOT: PSS/ ITO substrates. (Control: PEDOT: PSS, Target: PEDOT: PSS+ $3\text{ mg mL}^{-1} K_2SnO_3$). Target: K_2SnO_3 was incorporated into the PEDOT: PSS but not into the perovskite precursors. All films used for testing were top-capped with a PMMA thin film.

Supplementary Table S2. TRPL fitting data of Sn-Pb perovskite films deposited on PEDOT: PSS/ ITO substrates (Control: PEDOT: PSS, Target: PEDOT: PSS+ $3\text{ mg mL}^{-1} K_2SnO_3$).

Sample	A_1	τ_1 (ns)	A_2	τ_2 (ns)	τ_{avg} (ns)
Control (Top)	2620.3	949.7	544.0	2051.7	1139
Target (Top)	636.6	393.0	2326.1	1533.5	1285
Control(Bottom)	3280.4	736.0	44.7	4908.3	796
Target (Bottom)	1805.0	390.4	1102.4	1332.4	746

8, The authors think that the decreased PL and TRPL lifetime can be attributed to reduced non-radiative recombination. However, this is contradictory to many previous reported. The authors should measure and compare the PL and TRPL of perovskite films without and with the interface layer, respectively. Non-radiative recombination and carrier transport should be separately discussed. The unreasonable description should be deleted.

Response: We thank the reviewer for the valuable and perceptive suggestions.

In response, we have redesigned the experimental setup based on the reviewer's recommendations. As shown in Figures S19-20, we prepared perovskite films, with and without the PEDOT: PSS hole transport layer, and tested their steady-state PL and TRPL under excitation from both the top and bottom surfaces.

As shown in Figures S20a,c,e, for the perovskite film deposited on the ITO substrate, the target group exhibited stronger PL intensity and longer carrier lifetime, regardless of whether the excitation light was incident from the top or bottom. Specifically, the target group showed shorter fast decay (τ_1) and longer slow decay (τ_2), while the control group exhibited the opposite trend. In general, the slow decay is associated with radiative emission from a large number of perovskite excitons, while the fast decay is linked to non-radiative recombination (*Adv. Mater.*, 2018, 30(22): 1706924.). Therefore, the improvement observed in the target group can be attributed to the introduction of K_2SnO_3 , which promoted preferred orientation crystallization of the perovskite films, improved crystal quality, inhibited defect formation, and reduced non-radiative recombination channels.

For the perovskite films deposited on PEDOT: PSS HTL-coated substrates, when the excitation light was incident from the top surface, the PL intensity of the target group increased, and the carrier lifetime rose from 1139 ns to 1371 ns (Figures S20b, d, f). This improvement can be attributed to two major factors: 1) The introduction of K_2SnO_3 in PEDOT: PSS enhanced the wettability of the perovskite precursor solutions, leading to the formation of more uniform perovskite films. 2) The in-situ formation of $PbSnO_3$ seeds induced preferred orientation crystallization, enhancing the crystal quality of the films. When the excitation light was incident from the glass side, the target group exhibited stronger PL quenching, with the carrier lifetime decreasing from 796 ns to 700 ns. This reduction is attributed to the efficient hole carrier extraction and transport properties (*Joule*, 2024, 8, 2220-2237.) of PEDOT: PSS with K_2SnO_3 .

We have revised the manuscript accordingly, removing any misleading descriptions. Kindly refer to the revisions on Page 11 in our revised manuscript, where these changes have been incorporated. We again thank the reviewer for the valuable suggestions.

Supplementary Figure S20. (a-b) Schematic diagrams of PL measurements with excitation from the top and bottom of the films. (c) Steady-state and (e) time-resolved PL spectra of control and K₂SnO₃-treated Sn-Pb perovskite films deposited on ITO substrates. (d) Steady-state and (f) time-resolved PL spectra of control and target Sn-Pb perovskite films deposited on HTL-coated ITO substrates. Target: K₂SnO₃ was incorporated into both the perovskite precursors and PEDOT: PSS. All films used for testing were top-capped with a PMMA thin film.

Supplementary Table S3. TRPL fitting data of control and target Sn-Pb perovskite films deposited on ITO substrates.

Sample	A ₁	τ ₁ (ns)	A ₂	τ ₂ (ns)	τ _{avg} (ns)
Control (Top)	1351.4	832.1	1666.0	1499.6	1200
Target (Top)	756.0	510.1	2287.9	1779.7	1465
Control(Bottom)	1847.17	692.2	1074.64	1508.9	992
Target (Bottom)	677.7	381.9	2241.8	1442.1	1194

Supplementary Table S4. TRPL fitting data of control and target Sn-Pb perovskite films deposited on PEDOT: PSS-coated ITO substrates. For the target samples, K₂SnO₃ was incorporated into both the perovskite precursors and PEDOT: PSS.

Sample	A ₁	τ ₁ (ns)	A ₂	τ ₂ (ns)	τ _{avg} (ns)
Control (Top)	2620.3	949.7	544.04	2051.7	1139
Target (Top)	568.1	364.0	2306.5	1622.8	1371
Control(Bottom)	3280.4	736	44.7	4908.3	796
Target(Bottom)	561.6	344.4	2525.9	783.9	700

9, As one of the highlights is the device performance, the efficiencies of devices should be certified / third party verified.

Response: We sincerely thank the reviewer for the valuable suggestions.

In response, we have provided a certified report for one of our 2T tandem cells. The device achieved a certified reverse-scanned PCE of 27.60%, closely matching the lab-measured value of 28.2%, with a mismatch of approximately 2%. Due to the environmental sensitivity of Sn-Pb perovskites and the extended waiting time for third-party measurements, the device experienced slight degradation.

We appreciate the reviewer's understanding and thank the reviewer again for the constructive feedback.

Report No. 24TR060102

====Measurement Results====

	Forward Scan (Isc to Voc)	Reverse Scan (Voc to Isc)
Area	6.58 mm ²	
Isc	1.051 mA	1.068 mA
Voc	2.156 V	2.160 V
Pmax	1.757 mW	1.816 mW
Ipm	0.926 mA	0.960 mA
Vpm	1.896 V	1.892 V
FF	77.52 %	78.68 %
Eff	26.70 %	27.60 %

- Spectral Mismatch Factor $SMM_{top}=1.0055$, $SMM_{bot}=0.9887$.
- Designated Illumination area defined by a thin mask was measured by the measuring microscope.
- Test results listed in this measurement report refer exclusively to the mentioned measured sample.
- The results apply only at the time of the test, and do not imply future performance.

Supplementary Figure S31. Certification report for a representative all-perovskite tandem cell, incorporating a K_2SnO_3 -modified Sn-Pb subcell, issued by the Shanghai Institute of Microsystem and Information Technology (SIMIT).

Reviewer #2:

In this work, Ke et al. introduce a universal strategy for templating halide perovskite growth using oxide-based ABX_3 -structured seeds, demonstrating broad applicability across various perovskite compounds. While previous studies have explored seed materials, this work presents a unique approach by incorporating potassium stannate into the perovskite precursors to form $PbSnO_3$ as the seed material. These $PbSnO_3$ seeds provide notable advantages, including strong solvent stability and high lattice compatibility with perovskites. As a result, perovskite films templated with these seeds achieve high efficiency in both single-junction and tandem solar cells. This study presents intriguing findings of broad relevance to the community, and the results are well-articulated. I recommend its publication in Nature Communications after addressing the following minor revisions.

Response: We sincerely thank the reviewer for the thorough and careful reading of our manuscript, and we appreciate the constructive comments and much-valued suggestions.

1, Given that potassium stannate is introduced during perovskite film preparation, could the authors clarify the final distribution of potassium within the film? Please provide experimental evidence, such as EDS mapping, to illustrate potassium distribution in the perovskite layer.

Response: We thank the reviewer for the constructive comments.

In response, we conducted EDS measurements of the films. As shown in Figure S10, the K elements were distributed throughout the film. Additionally, the XRD patterns presented in Figure 2a revealed no shifts in peak positions for the corresponding perovskite thin film, indicating that K^+ ions did not incorporate into the perovskite lattice. Thus, we conclude that the K elements were primarily located on the film surface and at the grain boundaries.

Supplementary Figure S10. SEM image of a K_2SnO_3 -modified perovskite film with corresponding EDS elemental mapping, illustrating the distribution of key elements across the film.

2, The introduction of potassium stannate significantly enhances device performance, particularly the V_{oc} . Could the authors provide insight into the underlying mechanism that contributes to this V_{oc} improvement?

Response: We greatly appreciate the reviewer's insightful comment.

The increase in V_{oc} can primarily be attributed to significantly improved crystal quality and reduced carrier recombination. This is supported by several key measurements:

(1) The in-situ seed crystallization method effectively enhanced the crystal quality of the film, resulting in a preferentially oriented crystal structure dominated by the (100) crystal plane. This reduced film defects and facilitated carrier transport, contributing to the increase in V_{oc} .

(2) C - V test results (Figure 4f) revealed that the target device exhibited a higher V_{bi} (0.66 V) compared to the control device (0.51 V). This indicates that the addition of

K_2SnO_3 effectively enhanced V_{bi} , improving carrier extraction efficiency.

(3) PL and TRPL spectra (Figure S20) demonstrated that the introduction of K_2SnO_3 significantly increased PL intensity and extended carrier lifetime. Additionally, EIS results (Figure S25b) confirmed that K_2SnO_3 -modified devices exhibited larger recombination resistance, helping to suppress carrier recombination.

These results collectively suggest a substantial reduction in non-radiative recombination, leading to the observed increase in V_{oc} .

3, The reaction between potassium stannate and lead iodide yields lead stannate and potassium iodide as a by-product. Could potassium iodide also contribute positively to the film? Please discuss its potential influence within the perovskite films.

Response: Thank you for the excellent question.

Indeed, in addition to the PbSnO_3 seeds, the KI by-product also synergistically contributed to improving the film quality. Several studies have reported the positive effects of KI in enhancing device performance, as outlined below:

(1) Synergistic improvement of crystal quality: Studies have demonstrated that the introduction of KI enhances the defect density in perovskite films and reduces grain boundaries, leading to improved film quality (*Chem. Eng. J.*, 2023, 466: 142999.).

(2) Inhibition of ion migration and reduction of hysteresis effects: The strong bond energy between halide ions and K^+ effectively suppresses photoinduced ion migration at grain boundaries and layer interfaces in perovskite films. This inhibition reduces the hysteresis effects in perovskite devices and enhances their operational stability (*Nature*, 2018, 555(7697): 497-501.).

(3) Reduction of non-radiative recombination: The introduction of KI mitigates the formation and migration of vacancy defects, suppresses non-radiative recombination pathways, and extends carrier lifetimes (*ACS Appl. Mater. Interfaces*, 2020, 12(43): 48458-48466.).

These findings highlight the complementary role of KI in optimizing film quality and enhancing device performance.

4, In Figure 2e, SEM results reveal residual crystals at grain boundaries in the control group. Could the authors identify the composition of these residues and explain their formation? Additionally, some voids appear at the bottom of the target perovskite films. Could the authors discuss their origin?

Response: We speculate that the residuals at the grain boundaries may consist of $\text{PbI}_2/\text{SnI}_2$, consistent with results reported in the literature (*Nat. Energy*, 2024: 1-9.). To verify this, we introduced excess PbI_2 and SnI_2 into the perovskite precursor solutions and prepared corresponding perovskite thin films. As shown in Figure R6a, a large number of flake-like particles appeared at the grain boundaries of the films prepared with excess $\text{PbI}_2/\text{SnI}_2$. The corresponding XRD patterns (Figure R6b) confirmed the presence of excess $\text{PbI}_2/\text{SnI}_2$ in the final films. Based on these results, we conclude that the residual material at the grain boundaries is $\text{PbI}_2/\text{SnI}_2$.

Voids at the buried bottom interface:

During the crystal growth process, solvent evaporation occurs simultaneously.

However, the rapid crystallization of the Sn-Pb perovskite film prevents the solvent at the buried interface from escaping quickly, leading to incomplete crystal filling and the formation of voids or holes in the final films (*Science*, 2021, 373, 902-907; *Nat. Photonics*, 2023, 17(10): 856-864.).

Figure R6. (a) SEM images and (b) XRD patterns of perovskite films prepared by the control group, with excess PbI_2 , and with excess SnI_2 .

5, Potassium stannate was also introduced into the PEDOT hole transport layer to improve carrier transport. Could the authors elaborate on the rationale for this choice?

Response: We appreciate the reviewer's valuable comment. The introduction of K_2SnO_3 into the hole transport layer is mainly attributed to the following reasons:

(1) Improvement of perovskite precursor wettability: As shown in Figure S17, the introduction of K_2SnO_3 significantly reduced the contact angle of the perovskite precursor solution on PEDOT: PSS, indicating enhanced wettability. This promoted the formation of a more uniform film and reduced crystal defects at the buried interface.

(2) Improvement of conductivity: As shown in Figure S18ab, CAFM measurements demonstrated that the introduction of K_2SnO_3 increased the conductivity of the PEDOT: PSS film, from 481 pA in the control group to 728 pA. Additionally, J - V tests showed an increase in PEDOT: PSS film conductivity from $5.4 \times 10^{-4} \text{ S cm}^{-1}$ to $2.65 \times 10^{-3} \text{ S cm}^{-1}$ (Figure S18c).

(3) Reduction of non-radiative recombination: The introduction of K_2SnO_3 facilitated the rapid extraction of hole carriers. As shown in Figure S19, when excitation light was incident from the upper surface of the perovskite film, the target group exhibited stronger PL intensity and longer carrier lifetime due to the introduction of K_2SnO_3 in PEDOT: PSS. This improved the wettability of the perovskite precursor, promoted better crystallinity, and reduced non-radiative recombination. When excitation occurred from the bottom ITO glass side, the target film showed relatively weak PL intensity and shorter TRPL carrier lifetime due to rapid hole extraction and interface recombination. These effects are attributed to the enhanced conductivity of

PEDOT: PSS and improved carrier transport at the buried interface.

(4) Reduction of PEDOT: PSS acidity: The introduction of K_2SnO_3 adjusted the pH of the PEDOT: PSS solution from 2.51 to 4.87, reducing the acidity of PEDOT: PSS, helping minimize the erosion of the perovskite layer (Figure S16) and contributing to improved device stability.

Supplementary Figure S18. Conductive AFM images of PEDOT: PSS (a) without and (b) with K_2SnO_3 modification. (c) Electrical conductivity measurements of PEDOT: PSS and PEDOT: PSS+ 3 mg mL^{-1} K_2SnO_3 HTLs with a structure of ITO/HTL/Cu.

Supplementary Figure S19. Schematic diagram of PL measurements with excitation from the (a) top and (b) bottom of the films. (c) Steady-state and (d) time-resolved PL spectra of Sn-Pb perovskite films on PEDOT: PSS/ ITO substrates (Control: PEDOT: PSS, Target: PEDOT: PSS+ 3 mg mL^{-1} K_2SnO_3). All films used for testing were top-capped with a PMMA thin film.

Supplementary Figure S16. pH values of (a) a PEDOT: PSS solution and (b) a PEDOT: PSS solution with K_2SnO_3 (3 mg mL^{-1}).

6, In Figure 5c, please include the scale bar value in the figure legend for clarity.

Response: We appreciate the reviewer for pointing this out. The scale bar value of 600 nm has been added to the legend of Figure 5c.

7, It is noted that potassium stannate addition to the perovskite precursor leads to PbSnO_3 formation. Did the precursor solution become cloudy due to the insolubility of PbSnO_3 ? Additionally, could the authors comment on whether adding PbSnO_3 directly would yield similar results?

Response: We thank the reviewer for the valuable and perceptive suggestions.

As an inorganic compound, PbSnO_3 exhibits a relatively low solubility in DMF and DMSO. As shown in Figure R7a-b, when a small amount of K_2SnO_3 was added, the precursor solution remained clear, whereas excessive K_2SnO_3 led to turbidity and precipitation at the bottom of the bottle. In addition, as the amount of K_2SnO_3 added to the perovskite precursor increased, the solution became increasingly cloudy, and the Tyndall effect became more pronounced, indicating a higher density of aggregates in the precursors (*Nat. Synth.*, 2024. <https://doi.org/10.1038/s44160-024-00687-2>. *Angew. Chem. Int. Ed.*, 2024, 63(35): e202408586.). In contrast, for the direct addition of PbSnO_3 , as shown in Figure R7c-d, the solution was highly cloudy.

We directly compared devices prepared with K_2SnO_3 and PbSnO_3 -added precursors. As shown in Figure R8, the efficiency of the device with direct PbSnO_3 addition was 20.98%, which was very close to that of the control group (20.14%) and much lower than that of the device added with K_2SnO_3 (22.79%), further demonstrating the superior effectiveness of our K_2SnO_3 strategy.

Figure R7. Photographs of 0 mg mL⁻¹ (C), 0.5 mg mL⁻¹ (T₁), 2 mg mL⁻¹ (T₂) and 5 mg mL⁻¹ (T₃) K₂SnO₃ dissolved in perovskite precursor solutions under (a) natural light and (b) laser irradiation. The photos were taken after standing for 1h. (c) Photograph comparing 0 mg mL⁻¹ K₂SnO₃, 2 mg mL⁻¹ K₂SnO₃, and 2 mg mL⁻¹ PbSnO₃ added to perovskite precursor solutions. (d) Photograph of PbSnO₃ dissolved in mixed DMF and DMSO.

Figure R8. *J*-*V* curves of control, PbSnO₃-modified, and K₂SnO₃-modified single-junction Sn-Pb PSCs.